# The vacuolar iron transporter mediates iron detoxification in *Toxoplasma gondii*

Dana Aghabi[1], Megan Sloan [1], Grace Gill [1], Elena Hartmann[1], Olga Antipova[2], Zhicheng Dou [3], Alfredo J. Guerra[4,5], Vern B. Carruthers [4] & Clare R. Harding [1]✉

Iron is essential to cells as a cofactor in enzymes of respiration and replication, however without correct storage, iron leads to the formation of dangerous oxygen radicals. In yeast and plants, iron is transported into a membrane-bound vacuole by the vacuolar iron transporter (VIT). This transporter is conserved in the apicomplexan family of obligate intracellular parasites, including in *Toxoplasma gondii*. Here, we assess the role of VIT and iron storage in *T. gondii*. By deleting VIT, we find a slight growth defect in vitro, and iron hypersensitivity, confirming its essential role in parasite iron detoxification, which can be rescued by scavenging of oxygen radicals. We show VIT expression is regulated by iron at transcript and protein levels, and by altering VIT localization. In the absence of VIT, *T. gondii* responds by altering expression of iron metabolism genes and by increasing antioxidant protein catalase activity. We also show that iron detoxification has an important role both in parasite survival within macrophages and in virulence in a mouse model. Together, by demonstrating a critical role for VIT during iron detoxification in *T. gondii*, we reveal the importance of iron storage in the parasite and provide the first insight into the machinery involved.

The obligate intracellular parasite *Toxoplasma gondii* is an unusually promiscuous pathogen, able to infect most warm-blooded species, almost all nucleated cells and able to replicate in multiple tissue types. This lifestyle makes it an important pathogen of humans and livestock, where it is a major cause of miscarriage and economic loss[1]. Further, the indiscriminate lifestyle of *T. gondii* also suggests that the parasite can adapt to different environments to ensure a sufficient supply of essential nutrients. One key element required by the parasite is iron. Iron plays a central role in proteins essential for respiration in almost all living cells. In common with most eukaryotes, *T. gondii* contains iron[2] which is required for a number of essential cellular processes, including heme biosynthesis[3,4], iron sulphur (Fe-S) cluster biogenesis[5] and as a cofactor in several important proteins such as catalase[6,7], prolyl hydroxylases[8] and the deoxyribonucleotide synthesis enzyme ribonucleotide reductase.

Despite its essential role as a redox co-factor, the ease with which iron gains and loses electrons makes it potentially dangerous in the cellular environment[9,10]. Free iron reacts with $H_2O_2$, produced by aerobic respiration, via the Fenton reaction to form highly reactive hydroxyl and hydroperoxyl radicals (HO· and HOO·, collectively reactive oxygen species (ROS)) which can cause extensive damage to the cell. To avoid this fate, mammalian cells safely store most iron in the cytosol within protein cages of ferritin[11]. Ferritin has been highly conserved through evolution, with homologues found in plants and bacteria[11]. However, outside of the Metazoa, organisms including plants, yeast and protists transport iron into membrane-bound vacuoles through the action of an $H^+$-dependent antiporter, named CCC1 in yeast or the vacuolar iron transporter (VIT) in plants[12,13]. The apicomplexan phylum, including *T. gondii* and other key human and

[1]Wellcome Centre of Integrative Parasitology, School of Infection and Immunity, University of Glasgow, Glasgow, UK. [2]X-Ray Sciences Division, Advanced Photon Source, Argonne National Laboratory, Argonne, IL, USA. [3]Department of Biological Sciences, Clemson University, Clemson, SC, USA. [4]Department of Microbiology and Immunology, University of Michigan, Ann Arbor, MI, USA. [5]Present address: Cayman Chemical Company, Ann Arbor, MI, USA. ✉e-mail: clare.harding@glasgow.ac.uk

veterinary parasites such as *Plasmodium* spp. (causative agent of malaria) lack ferritins entirely and presumably rely on storage of iron within a membrane bound compartment. In support of this hypothesis, a homologue of the plant vacuolar iron transporter was described in *Plasmodium* and shown to be a functional iron transporter that complemented a Δccc1 yeast strain[14–16]. Deletion of VIT in *Plasmodium* demonstrated that the gene was essential for replication in both blood and liver stage parasites. Further, the absence of VIT rendered *Plasmodium* more susceptible to iron overload and more resistant to iron removal[16].

However, *Plasmodium* spp. undergo their asexual lifecycles in the unusual environment of the red blood cell, with very high levels of iron in the form of haemoglobin. In contrast, the ubiquitous pathogen *T. gondii* is a generalist and so faces different iron stresses throughout its lifecycle. We were interested in how *T. gondii* stores iron, and the effects disrupting iron storage has on the parasite. We show that iron is localised within a compartment of the parasite, distinct from zinc, supporting the hypothesis that iron is stored in a vacuole. By knocking out VIT, we find that it is not essential for replication, however, expression of VIT confers a survival advantage during in vitro culture. Tagging demonstrated VIT has a highly dynamic localisation throughout the cell cycle in *T. gondii*. We see transient colocalisation with markers of the plant like-vacuolar compartment (PLVAC), suggesting this as the location of stored iron. We show that deletion of VIT makes the parasites significantly more sensitive to high levels of exogenous iron, and that this effect is mediated through an increase in reactive oxygen species (ROS) upon the addition of iron. Further, VIT expression is regulated in response to changes in exogenous iron levels. We also find that parasites lacking VIT are more susceptible to killing by activated macrophages, and this is likely a key factor in their reduced virulence in mice.

## Results

### Deletion of VIT leads to a moderate growth defect

The presence of a VIT homologue in *T. gondii* suggests that iron is stored in a membrane-bound compartment. To examine this, we looked at iron localisation by X-ray fluorescence microscopy (XFM). We found that iron in extracellular and recently invaded parasites localised to a distinct region of the cell, which showed limited overlap with either zinc or calcium (Fig. 1a). Instead, zinc, calcium and phosphorus appeared to closely overlap, supporting previous work proposing that zinc is stored in compartments with similarities to acidocalcisomes[17,18]. This also complements previous work in *Plasmodium* showing that zinc and iron appear to be stored in separate compartments[19]. VIT from *T. gondii* was previously identified through homology to plant and *Plasmodium* transporters[16]. To confirm the conservation of key residues, we aligned the protein sequence of *T. gondii* VIT (TGGT1_266800) with sequences from *Plasmodium*, yeast (CCC1) and plants (*Arabidopsis thaliana* and *Eucalyptus grandis*, as the crystal structure of EgVIT was recently published[20]) (Fig. 1b and Fig. S1). The *T. gondii* VIT had 33% identity with AtVIT1, including conservation of key residues D43 and M80, recently shown to be essential for iron binding[20]. To investigate the function of VIT, the coding region was amplified from *T. gondii* cDNA and cloned into a constitutive yeast expression vector and expressed in wild type (BY4742) and Δccc1[21] *S. cerevisiae* (Fig. S2). Expression of full-length VIT in wild type yeast appeared toxic, with fewer and smaller colonies. We also tested an N-terminal truncation (sVIT$_{63-313}$) as this truncation of PfVIT was shown to successfully complement a Δccc1 mutant[16], however, sVIT$_{63-313}$ also showed toxicity in wild type yeast. The toxicity of both VIT constructs appeared abrogated in the Δccc1 line, however, we did not observe any complementation of the iron-hypersensitivity phenotype. This contrasts with *Plasmodium* VIT[16] and may be due to differences in expression or codon usage. To investigate VIT in the native context, the open reading frame of VIT in *T. gondii* was deleted using two small

guide RNAs targeting the 5′ and 3′ regions of the open reading frame (see Table 1) and replaced with a cassette encoding mNeonGreen under the control of the SAG1 promoter (Fig. 1c) in a ΔKu80 parental cell line. Integration of the cassette and loss of the native ORF was confirmed by PCR (Fig. 1d). ΔVIT parasites were viable in culture and able to form cleared areas on a monolayer of host cells at 6 days post infection (Fig. 1e). A genome-wide essentiality screen predicted that loss of VIT had a mild effect on parasite growth (phenotype score −1.22)[22]. To examine the growth of the parasites in more detail, we quantified the number of parasites within a parasitophorous vacuole at 24 h post infection (Fig. 1f). The ΔVIT line had fewer parasites per vacuole (significant at the 2-cell, $p = 0.0005$ and the 8-cell stage, $p = 0.0037$, multiple $t$ tests corrected by Holm-Sidak), demonstrating that VIT has a role in parasite replication under normal growth conditions. We also examined the parasite's ability to survive extracellularly. Mechanically released parasites were incubated extracellularly at 37 °C for the indicated time, then allowed to invade fresh host cells. We then quantified the number of plaques formed after 6 days. We found no difference in extracellular survival between the parental and ΔVIT parasites ($p > 0.05$, multiple $t$ tests corrected by Holm-Sidak) (Fig. 1g), demonstrating that VIT does not have a role in parasite survival outside of the host cell.

As an iron transporter, it could be expected that deletion of VIT would lead to a change in the amount and distribution of iron within the cell. Using XFM we examined iron distribution in ΔVIT parasites shortly after invasion. Although the signal was heterogeneous, we generally saw less iron within cells compared to the parental line. Some cells appeared to have no or very little iron while in other cells we saw a more fragmented signal which did not overlap with the other elements detected (Fig. 1h). Overall, we determined that VIT has a small role in parasite replication and that it may have a role in iron distribution within the cells.

### ΔVIT parasites are hypersensitive to excess iron and have lower cellular iron levels

To confirm the role of VIT in *T. gondii*, we complemented the ΔVIT line with VIT cDNA under the native promoter by inserting *pvit-vit-3′dhfr* into the uracil phosphoribosyltransferase *(uprt)* locus and selecting with 5-flurodeoxyuridine (FUDR) (Fig. S3a). Integration of the cassette and disruption of the locus was confirmed by PCR (Fig. S3b) and VIT expression in the ΔVIT + VIT line was confirmed by qPCR (Fig. S3c). Interestingly, this was around double the wild type level, possibly due to the change in genomic context or altered 3′ UTR. In both yeast and *Plasmodium*, deletion of *Sc*CCC1 or *Pb*VIT leads to hypersensitivity to excess iron[12,16]. To examine this in *T. gondii*, we performed a plaque assay under increasing concentrations of ferric ammonium chloride (FAC). We found that ΔVIT parasites were more sensitive to excess FAC, forming fewer and significantly smaller plaques ($p < 0.001$ at 200 μM FAC, two way ANOVA) than the parental line (Fig. 2a–c) and this phenotype could be somewhat rescued in the ΔVIT + VIT line. Interestingly, we did not see complete rescue of the phenotypes in our complemented line, potentially linked to the increased expression of *vit*. To confirm that deletion of VIT leads to iron hypersensitivity, we performed a competition assay under standard conditions and with excess FAC. To serve as controls, two new lines were constructed in RhΔHXPGRT, where the coding region of *ku80* was replaced with a cassette expressing mNeonGreen (ΔKu80::mNeonGreen, called mNeon) or tdTomato (Fig. S3d), resulting in two lines which expressed fluorescent proteins in place of the endogenous *ku80* gene. We confirmed that these lines were fluorescent using flow cytometry and the loss of the *ku80* gene by PCR (Fig. S3e). To assess the fitness defect of the ΔVIT strain, ΔVIT or mNeon parasites were mixed with tdTomato in an equal ratio, and cultured with, or without, 200 μM FAC. The proportion of green parasites in the population was assessed at each passage by flow cytometry. The proportion of mNeon parasites did not

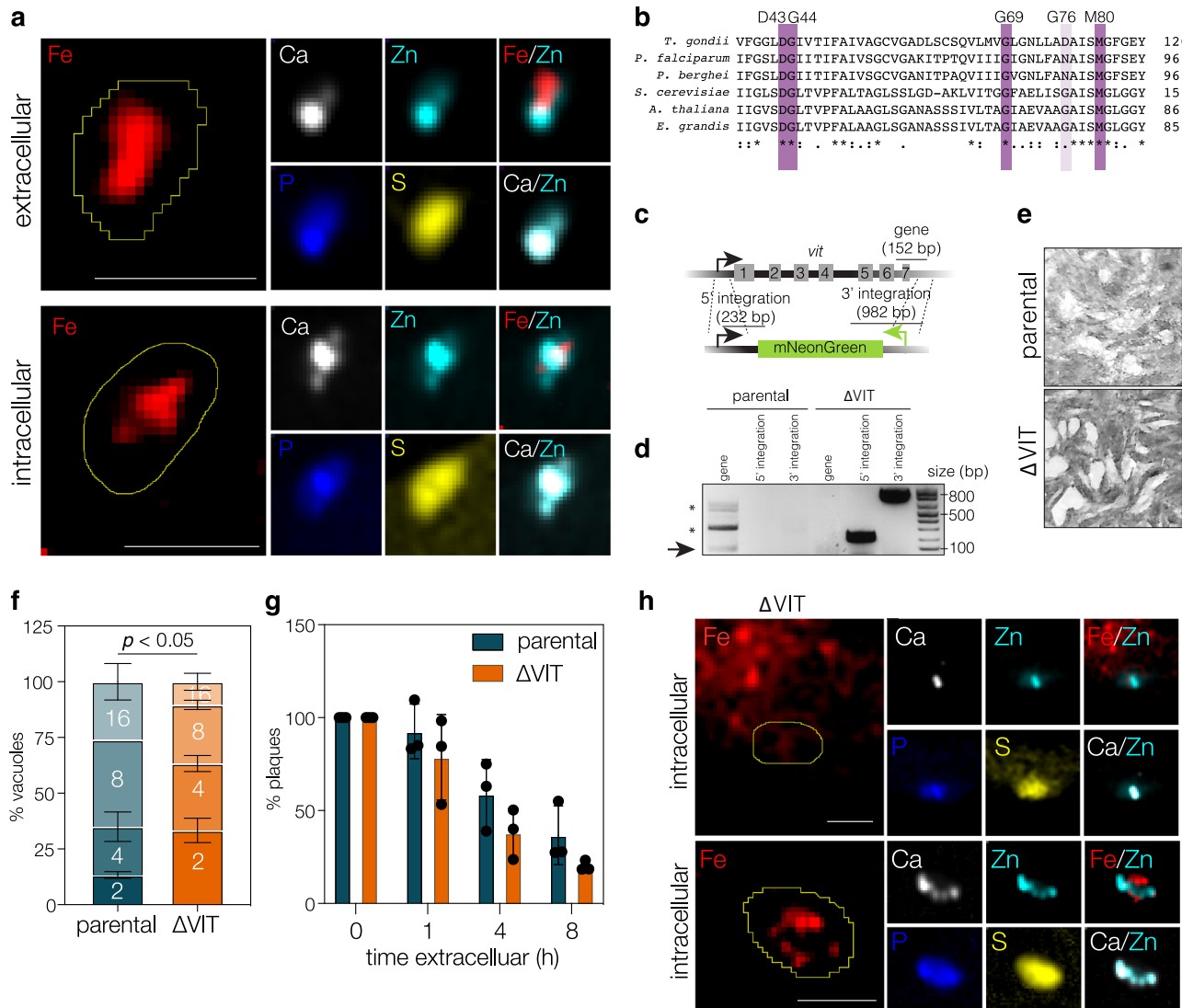

**Fig. 1 | Iron localisation and construction and analysis of a ΔVIT parasite line.**
**a** X-ray fluorescence microscopy of extracellular and intracellular *T. gondii*. Limited overlap between Fe and Zn was observed, however overlap between Zn, Ca and P was observed in both intracellular and extracellular parasites. Numbers indicate Pearson's correlation between channels. Line indicates the outline of the parasites. Representative of 3-4 parasites. Scale bar 5 μm. **b** Alignment of VIT using Clustal Omega from *T. gondii* (TGGT1_266800), *P. falciparum* (PF3D7_1223700), *P. berghei* (PBANKA_1438600), *S. cerevisiae* (ScCCC1), *A. thaliana* (AtVIT1) *and E. grandis* (EgVIT1). Identity (*) and similarity (.,:) indicated. Key conserved residues for iron binding highlighted in purple. **c** Schematic indicating the method used to replace the endogenous *vit* gene with the mNeonGreen cassette. **d** PCR reactions confirming the replacement of the endogenous gene, expected sizes indicated on

schematic. *represents secondary bands, possibly unspecific. **e** Plaque assay demonstrating that the ΔVIT line is viable. Representative of two independent experiments. **f** Quantification of number of parasites/vacuole at 24 h post infection. Results are the mean of $n = 3$ independent biological replicates, ±SD. $p$ value from multiple two tailed $t$ tests, corrected by Holm-Sidak, $p = 0.0005$ at 2-cell stage and $p = 0.0037$ at 8-cell stage. **g** Quantification of extracellular survival, normalised to 0 h. Bars are the mean of $n = 3$ independent biological replicates, ±SD. **h** X-ray fluorescence microscopy examining elemental composition of intracellular ΔVIT parasites. No change was seen in Zn, Ca, P or S in ΔVIT cells compared to intracellular parental parasites, however, Fe appeared potentially mislocalised. Numbers indicate Pearson's correlation between channels. Line indicates the outline of the parasites. Representative of 3–4 parasites. Scale bar 5 μm.

change over the experiment, however the ΔVIT line was significantly ($p = 0.002$, $t$ test, Holm-Sidak corrected) outcompeted by 3 days post infection (Fig. 2d). Addition of FAC exacerbated this phenotype, ΔVIT parasites were significantly ($p = 0.001$, $t$ test, Holm-Sidak corrected) outcompeted by 2 days post infection in the presence of excess iron and were almost undetectable by 4 days post infection. This shows that VIT has a role in growth under normal conditions, and lack of VIT sensitises parasites to exogenous iron. To confirm this result and provide extra confidence in our genetic manipulation, we constructed a second ΔVIT strain by replacing the *vit* gene with a DHFR cassette (Fig. S4a), named ΔVIT::DHFR$_{TS}$. We confirmed integration by PCR (Fig. S4b) and confirmed these parasites were also hypersensitive to excess iron by plaque assay (Fig. S4c, d).

To quantify the iron hypersensitivity of the ΔVIT parasites, we infected host cells in 96-well plates with mNeon or ΔVIT parasites and treated with increasing concentrations of FAC for 4 days before measuring the fluorescence of each well using a plate reader, normalised to untreated wells (Fig. 2e). This allowed us to quantify the degree of parasite proliferation in the presence of a range of iron concentrations. FAC treatment significantly ($p < 0.0001$, extra sum of squares F-test) inhibited ΔVIT parasite proliferation compared to the mNeon strain, with an EC$_{50}$ of 0.184 μM (95% CI: 0.08 to 0.408 μM), compared to the mNeon EC$_{50}$ of 61 μM (95% CI: 21.1 to 177.7 μM). This hypersensitivity was partially rescued in the complemented strain (EC$_{50}$ of 6.998 μM (95% CI: 1.81 to 24.7 μM), significantly higher than the ΔVIT strain ($p < 0.0001$, extra sum of squares $F$-test) but still

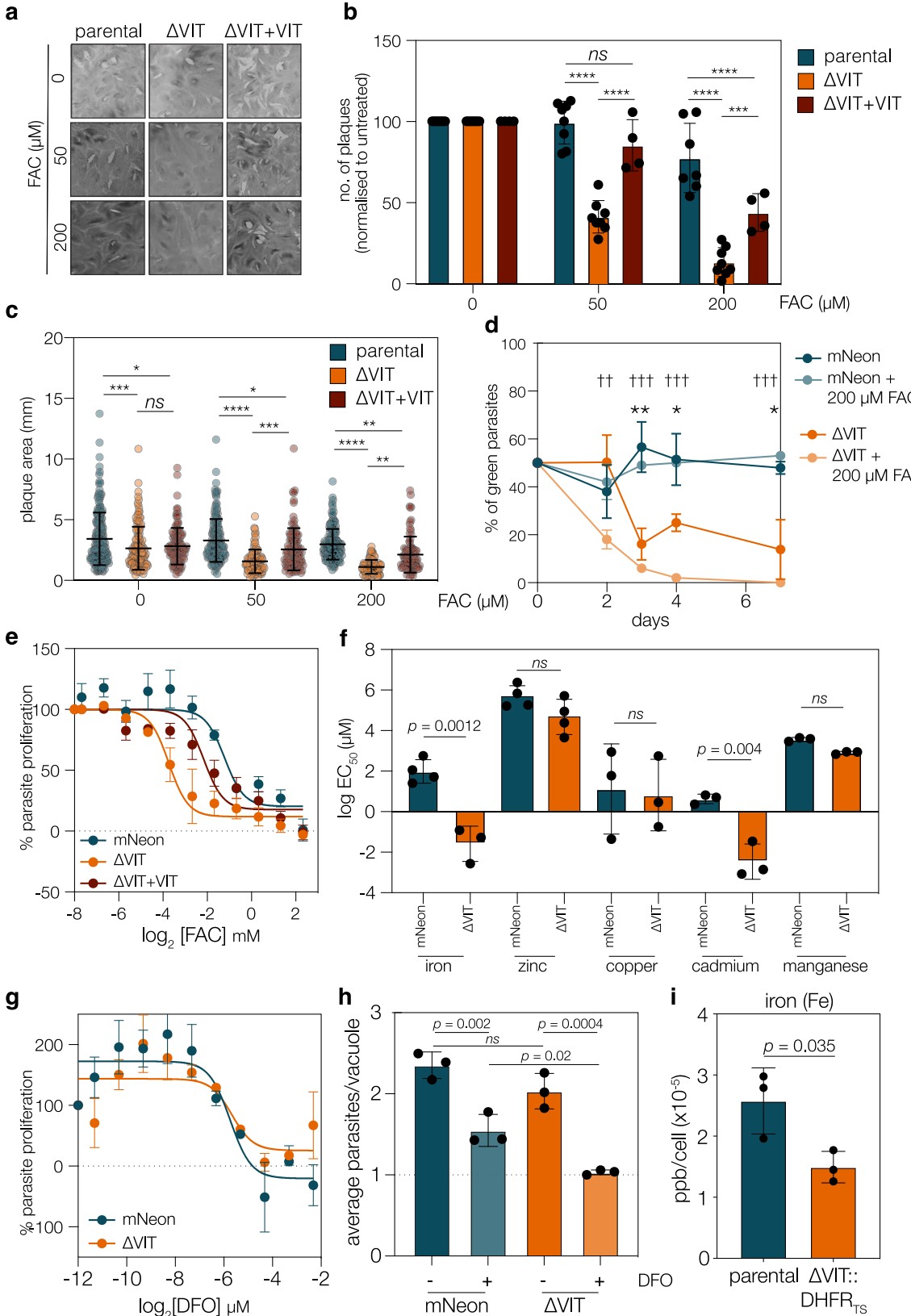

10-fold lower than the mNeon control line, as seen in the plaque assays above.

VIT homologues from various species have been shown to transport other metals, including manganese[12] and zinc[23]. To test if deletion of VIT altered the sensitivity of *T. gondii* to metals other than iron, we treated parasites as above with increasing concentrations of zinc,

copper, cadmium, and manganese and determined the EC$_{50}$. We saw that the absence of VIT did not affect the response to treatment with excess zinc, copper, or manganese (*t* test, *p* values > 0.05) (Fig. 2f and Fig. S5a–e). While this is not direct evidence that VIT from *T. gondii* is unable to transport these substrates, it does show that VIT is not required for the detoxification of the selected metals. However, we did

**Fig. 2 | ΔVIT parasites are more susceptible to iron overload. a** Plaque assay with indicated concentrations of FAC (ferric ammonium chloride). ΔVIT parasites more susceptibile to exogenous iron, somewhat complemented by re-expression of VIT. **b** Quantification of number of plaques (normalised to the untreated) for parental, ΔVIT and ΔVIT + VIT parasites after 0, 50 or 200 μM FAC treatment. Results from 3 (ΔVIT + VIT), 6 (ΔVIT) or 8 (parental) independent experiments, bar at mean ± SD, $p$ values from two-way ANOVA, Tukey corrected for multiple comparisons. ***$p = 0.0002$, ****$p < 0.0001$. **c** Plaque area for experiments above, $p$ values from one-way ANOVA, Tukey corrected for multiple comparisons **d** tdTomato parasites were mixed with mNeon and ΔVIT parasites in a 1:1 ratio and untreated or treated with 200 μM FAC. ΔVIT parasites were significantly outcompeted by 3 days post infection, or 2 days in the presence of excess iron. Points are the mean of $n = 4$, ±SD. **$p = 0.001$ **$p = 0.003$, ***$p = 0.0006$, $t$ test between mNeon and ΔVIT, †† $p = 0.001$, ††† $p < 0.0001$, $t$ test between mNeon +200 μM FAC and ΔVIT + 200 μM FAC, all two

tailed $t$ test, corrected for multiple comparisons by two-stage step-up (Benjamini, Krieger, and Yekutieli) **e** Dose-response curve showing ΔVIT parasites more sensitive to FAC that the parental, complemented line shows partial rescue. Points are the mean of $n = 4$ (mNeon) or 3 (ΔVIT and ΔVIT + VIT) independent experiments, ± SEM. **f** Graph showing mean EC$_{50}$ for indicated metals for mNeon and ΔVIT parasites, each point represents a biological replicate, performed in triplicate. Bars at mean, ±SD. $p$ values from two tailed $t$ test. **g** ΔVIT parasites did not show any significant change in the EC$_{50}$ upon DFO treatment. Results are $n = 3$, ± SEM. **h** mNeon and ΔVIT parasites were allowed to invade HFF cells untreated, or pre-treated with DFO for 24 h. At 14 h post invasion, average parasite/vacuole were quantified. Results mean of $n = 3 ±$ SD, at least 100 vacuoles counted/experiment. $p$ values from one way ANOVA with Holm-Sidak correction. **i** ICP-MS quantification from parental and ΔVIT::DHFR$_{TS}$ parasites. Bars are at the mean of $n = 3$, ±SD, $p$ value from two tailed $t$ test.

see a significant shift in the sensitivity to the heavy metal cadmium (EC$_{50}$ of parental line 3.41 μM (95% CI: 1.63 to 6.34 μM), EC$_{50}$ of ΔVIT 0.04 μM (95% CI: 0.008 to 0.149 μM), $p = 0.0009$, extra sum of squares $F$-test) (Fig. 2f and Fig. S5d). Cadmium has complex effects on mammalian cells and tissues[24] and there is evidence that cadmium can lead to increased ROS production[25]. To determine if the ΔVIT parasites were more susceptible to general increases in oxidative stress, we treated the parasites with sodium arsenite (Ars), which induces oxidative stress through a number of mechanisms[26] and has been successfully used in *T. gondii* to induce oxidative stress[27]. The concentrations of Ars used did not affect host cell viability as measured by Alamar blue assay (Fig. S5f). We saw no significant change in the sensitivity between parental line and the ΔVIT parasites to Ars treatment (Fig. S5g), demonstrating that the effect of VIT on cadmium toxicity is somewhat specific, and that ΔVIT parasites do not have a general increased sensitivity to oxidative stress. VIT from the distantly related diatom *Phaeodactylum tricornutum* is transcriptionally regulated by cadmium[28] where the authors suggest that Cd$^{2+}$ may also be stored in a vacuole. Responses to cadmium have not previously been examined in the Apicomplexa, however, our results here may suggest a link between iron and cadmium detoxification in *T. gondii*.

In *Plasmodium*, deletion of VIT also led to a change in the sensitivity to the iron chelator deferoxamine (DFO)[16]. However, in *T. gondii* we saw no change in the ability of the parasites to replicate in the presence of increasing concentrations of iron chelator (Fig. 2g) (EC$_{50}$ of mNeon 160 μM (95% CI: 65.9 to 362.2 μM) and ΔVIT 101.3 μM (95% CI: 44.9 to 233.5 μM), $p > 0.05$, extra sum of squares $F$-test) over the 4 days of the experiment. If VIT is responsible for storing iron, it is possible that in its absence, parasites would be disadvantaged in replication immediately after moving to an iron-poor environment. To simulate this, we pretreated host cells with DFO for 24 h to deplete iron, before adding mNeon or ΔVIT parasites and fixed after only 14 h post invasion. We then quantified the average number of parasites per vacuole. We found that pretreatment with DFO significantly reduced the average parasites/vacuole in both the mNeon and the ΔVIT parasite lines (Fig. 2h), demonstrating the importance of iron to parasite replication. Although we do see a replication defect in the ΔVIT strain (Fig. 1f), at this early point in infection it was not significant. However, in an iron-poor environment there was a significant ($p = 0.02$, one way ANOVA with Holm-Sidak correction) decrease in the average parasite/ vacuole in the ΔVIT strain compared to the mNeon strain, suggesting that the potential disruption of iron storage inhibits the ability of parasites to initiate replication in an iron-depleted environment.

If VIT binds iron and has a role in mediating parasite survival under high and low iron conditions, it is possible that its loss affects iron levels in the parasite. The above results suggested a loss of stored iron. To assess if iron levels changed in the ΔVIT::DHFR$_{TS}$ parasites, we quantified total parasite-associated iron using inductively coupled plasma-mass spectrometry (ICP-MS). Using this method, we found a mean of $2.6 \times 10^{-5}$ ppb of Fe/parasite in the parental parasite line

(Fig. 2i). In the absence of VIT, we saw a significant ($p = 0.035$, $t$ test) reduction in iron to approximately $1.5 \times 10^{-5}$ ppb/parasite, approximately 60% of the parental level. As a control, we saw no change in the levels of zinc between the parasite lines (Fig. S5h). A reduction in total iron has been observed before upon disruption of vacuolar iron transporters in yeast and plants[12,29], and supports the role of VIT in *T. gondii* iron storage.

As a putative iron transporter, we predict that VIT will bind directly to iron. Due to the transmembrane helices and the highly reactive nature of iron, this is difficult to assess directly in vitro. However, many proteins, including transporters, change in their tendency to aggregate at high temperatures when bound to their ligand, and this can be assessed using a cellular thermal shift assay[30-32] (CETSA) which was recently successfully used in *T. gondii* to assess the interaction of the kinase CDPK1 to a small molecule[33]. We endogenously tagged VIT at the C-terminal using 3xHA tags and confirmed that it was detectable as a single band at the expected size of 35 kDa (Fig. S3f). To determine the thermal stability, cells were untreated or treated with 2 mM FAC, lysed and subjected to a range of temperatures (37–64 °C) and the soluble fractions probed by western blot. The kinase CPDK1 was used as a control as it has been investigated by CETSA[33] and has the ability to bind to the divalent cation Ca$^{2+}$ but is not expected to bind to iron. As expected, incubation with FAC did not affect the solubility of CDPK1, which showed 50% protein solubility at 47.7 °C, comparable to the previously published inflection point of 47.2 °C[33]. Alternatively, VIT-HA showed significantly ($p = 0.028$, Extra sum of squares $F$-test) reduced protein stability upon FAC treatment (Fig. S3g). While not conclusive evidence that VIT binds iron, this change in thermal stability provides additional supporting data that VIT is modified in structure in the presence of iron.

## VIT has a dynamic localisation throughout the lytic cycle

In *Plasmodium*, VIT colocalized with markers of the endoplasmic reticulum (ER) throughout the parasite's lifecycle[16]. To determine the localisation of VIT in *T. gondii*, we used our endogenously tagged VIT-HA line to examine the localisation of the protein. Upon IFA, we observed VIT-HA expression as a single point in extracellular parasites or within an hour post invasion. This point appeared to fragment between 1–6 h post invasion and by 24 h post infection VIT-HA signal was seen in multiple small structures throughout the cytoplasm of the parasite (Fig. 3a). We quantified this fragmentation using an automated ImageJ macro and saw a significant increase ($p < 0.0001$, one way ANOVA with Tukey correction) in the number of VIT-HA foci per parasite between 1 and 6 h post infection (Fig. 3b). To confirm this localisation, we also endogenously tagged VIT with 3xmyc and saw a similar dynamic localisation through the cell cycle (Fig. S6a). To determine if changes in the localisation were associated with a change in abundance, we also assessed the level of VIT-HA by western blot and found no significant change in protein abundance (Fig. 3c, d). Upon

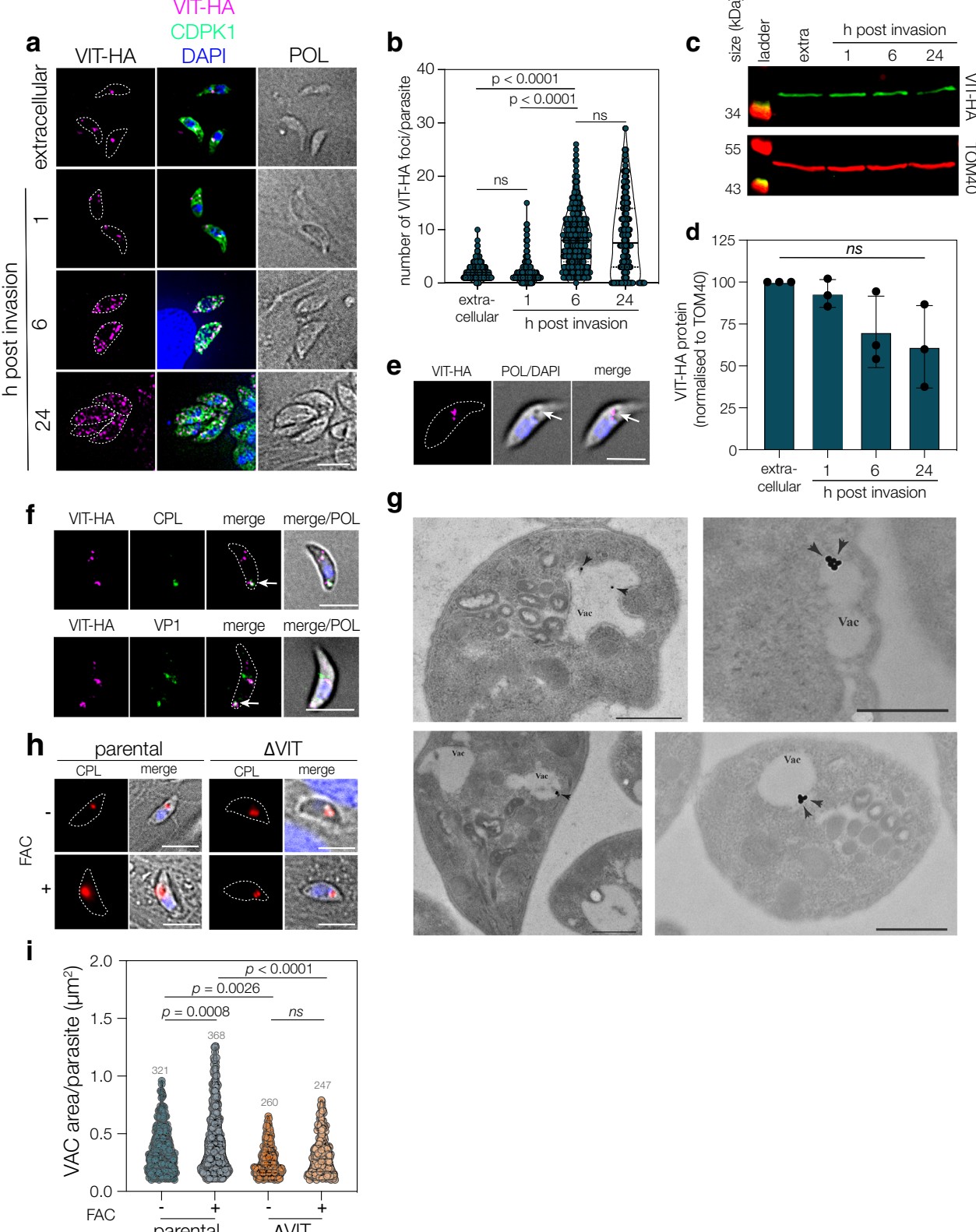

imaging extracellular parasites, we observed that the VIT-HA foci frequently overlapped with a vacuolar structure visible in phase contrast (Fig. 3e, arrows). This structure is reminiscent of the plant-like vacuolar compartment (PLVAC)[34]. To investigate if VIT-HA was localised to the PLVAC, we co-stained extracellular VIT-HA parasites with the PLVAC-markers CPL[35] and VP1[36]. In both cases we saw partial overlap between VIT-HA and the PLVAC markers (Fig. 3f, arrows),

suggesting multiple sub-populations of VIT-HA positive structures. To confirm that the HA tag did not alter VIT localisation, we also saw overlap between VIT-myc and CPL (Fig. S6b), confirming that VIT is frequently associated with this compartment. As a further test, we transiently overexpressed CRT-GFP and CRT-mCherry, both previously associated with the PLVAC[34,37], however, did not see any overlap in extracellular parasites (Fig. S6c).

**Fig. 3 | VIT has a dynamic localisation which alters through the lytic cycle. a** IFA of VIT-HA showing a dynamic localisation through the lytic cycle, VIT-HA exists at a single point in extracellular and 1 h post-invasion parasites before fragmenting at 6–24 h post invasion to a few small foci throughout the parasite. POL polarised light. Scale bar 5 µm. **b** Violin plot of number of foci/parasite from 3 independent replicates, 100 parasites/replicate. Bars at mean and quartiles. *p* values from one way ANOVA with Holm-Sidak correction. **c** Western blot showing VIT-HA at indicated time points, TOM40 used as a loading control. **d** Quantification of VIT-HA levels from three independent experiments, bar at mean ± SD, ns no significant change. **e** Extracellular VIT-HA parasites demonstrating co-localisation of VIT-HA with a vacuole visible by phase contrast (arrow). Representative of two independent experiments. Scale bar 5 µm. **f** VIT-HA overlaps with the PLVAC markers CPL and VP1 in extracellular parasites. Representative of three independent experiments. **g** Extracellular VIT-HA parasites imaged by immunoelectron microscopy. VIT-HA signal was frequently seen at the vacuole (VAC). Scale bar 500 nm. Representative of two independent experiments. **h** Parental and ΔVIT parasites, allowed to invade and treated for 1 h with excess FAC, were stained with anti-CPL, a marker for the VAC. Parental, but not ΔVIT parasites showed a swelling of the VAC. Scale bar 5 µm. **i** Violin plot of area of the PLVAC (as assessed by CPL staining) from the above experiment. Results of two independent replicates, *n* parasites indicated above plot, bar at mean ± quartiles. *p* values from one way ANOVA with Holm-Sidak correction.

To investigate the localisation more fully, we imaged extracellular parasites with immunoelectron microscopy against the HA tag. We found VIT-HA signal at the PLVAC (Fig. 3g) in two independent experiments, confirming our predicted localisation. If VIT localises to the PLVAC, it is possible that the PLVAC is the site of iron storage in the cell. To determine how the PLVAC responded to excess iron, we treated parasites with high levels of iron (2.5 mM FAC) to be sure of overwhelming any host cell buffering effects and determined the area of the PLVAC in an unbiased manner using an automated macro and the luminal PLVAC marker CPL. We found that the total area of CPL signal significantly increased (Fig. 3h, i) when parental parasites were treated with high exogenous iron ($p = 0.0008$, one way ANOVA with Dunnett's correction). We saw a small change in PLVAC area between the parental and ΔVIT parasites under normal conditions, but interestingly the area of the PLVAC did not change when we treated ΔVIT parasites with excess iron (Fig. 3h). Although we cannot rule out the role of other cellular changes, such as the redox state of the cell on PLVAC morphology, these results suggest the PLVAC is altered by excess iron through the actions of VIT, providing supporting evidence that VIT functions in transporting iron across the vacuolar membrane in *T. gondii*.

## VIT expression is regulated by iron availability

The expression, localisation and activity of transporters are frequently regulated by the availability of the substrate. This has been well established in model organisms[13,38–40] and was recently reported for the arginine transporter ApiAT1 in *T. gondii*[41]. To determine if iron affects expression of VIT, we performed qRT-PCR to assess the transcription of VIT under differential iron conditions. Unexpectedly, we found that transcript levels were significantly decreased ($p = 0.007$, one sample *t* test) (Fig. 4a) at 24 h upon treatment with excess iron. This is dissimilar to yeast, where excess iron upregulates CCC1 at the transcript and protein levels[42]. In contrast, removal of iron by DFO treatment did not lead to a significant change in *vit* RNA levels. As transcript and protein levels do not always correlate, we also examined changes at the protein level by western blotting using VIT-HA. We found that treatment with FAC also led to a variable but significant ($p = 0.043$, one sample *t* test) decrease in VIT protein levels (Fig. 4b, c), however interestingly, removal of iron by DFO also led to a decrease ($p = 0.032$, one sample *t* test) in VIT-HA levels (normalised to the mitochondrial protein TOM40) despite no change in RNA levels. As well as regulation at the mRNA and protein level, the function of transporters can also be regulated by changes in localisation[38]. We assessed the localisation of VIT-HA upon changes in iron levels. As described above, we quantified the number of VIT-foci after treatment with FAC or DFO at 6 and 24 h post infection. At 6 h post infection we saw a significant ($p = 0.0004$, one way ANOVA with Tukey) decrease in the number of foci/parasite upon treatment with DFO but not FAC (Fig. 4d). By 24 h post infection we saw a significant decrease in foci number under both conditions ($p < 0.0001$, one way ANOVA with Tukey) (Fig. 4e). This may show a different response to high or low iron, or may reflect the speed at which the treatments take effect within the cells. In either case, changes in iron levels appear to alter the transcription, protein level and localisation of VIT within the parasite.

## Transcriptional response to the absence of VIT

The finding that VIT appears regulated by iron led us to examine how the transcriptome changes in the absence of VIT and correct iron storage. RNAseq was performed on the parental and ΔVIT strains in triplicate. We confirmed that none of the reads mapped to the coding region of VIT in the ΔVIT strain, validating our KO (Fig. S7a). Under these conditions, 70 genes were downregulated upon deletion of VIT ($\log_2$ fold change (LFC) < −1, adjusted *p* value (Padj) < 0.05) and 62 were upregulated (LFC > 1, adjusted $p < 0.05$) under standard growth conditions (Fig. 4f, Supplementary Data 1). We saw significant upregulation of a potential plasma membrane ABC-type transporter (TGME49_239020). This gene had strong structural homology ($E = 7.1$ $e^{-135}$, HHPRED[43]) to multidrug efflux transporters and is named multidrug resistance protein 1 (MDR1) in *Plasmodium*[44]. Interestingly, work in mammalian cancer cells has shown that removal of iron led to a decrease in MDR1 levels[45] and disrupting iron storage through increased expression of ferritin led to an increase in MDR expression[46]. Upregulation of this transporter may thus be an adaptation by the cell to attempt to remove excess iron from the parasite, in the absence of correct storage.

There are two well studied pathways involving iron in *T. gondii*, the heme biosynthesis pathway and Fe-S cluster biogenesis. The contribution of VIT-based iron storage to these pathways is currently unknown. We examined the transcriptional changes in the heme biosynthesis pathways (Fig. S7b) and found there was a general upregulation of several components of the pathway upon VIT deletion, most significantly PBGD. The only gene in this pathway which directly interacts with iron, ferrochelatase, was not altered, however, there was a significant upregulation in a cytochrome c heme lyase, an important destination for heme in the parasite. *Toxoplasma* encodes three iron-sulfur cluster biogenesis pathways, localised to the cytosol, mitochondrion and apicoplast. Interestingly, we saw opposing changes in the mitochondrial ISC pathway, with downregulation of the scaffold IscU and upregulation of the acceptor IscA (Fig. S7c). In both mammalian and yeast cells, IscU expression is iron dependent, suggesting that the altered iron storage in the ΔVIT parasites is altering iron-dependent processes within the cell[47,48]. Overall, we saw significant changes in transcription upon deletion of VIT, with changes in transcription of genes and pathways known to require iron.

## Iron overload leads to ROS accumulation in the absence of VIT

Ferrous iron can lead to the production of reactive oxygen species (ROS) through the Fenton reaction[9] and loss of CCC1 in yeast is associated with increased oxidative stress[49]. To determine if absence of VIT affected ROS accumulation in *T. gondii*, we stained parasites with CellROX Deep Red, which fluoresces in the presence of ROS, and quantified fluorescence by flow cytometry. In the parental line, treatment with high concentrations of FAC (2 mM) did not significantly increase the ROS levels, suggesting the presence of effective mechanisms for controlling iron toxicity in these cells. There was no

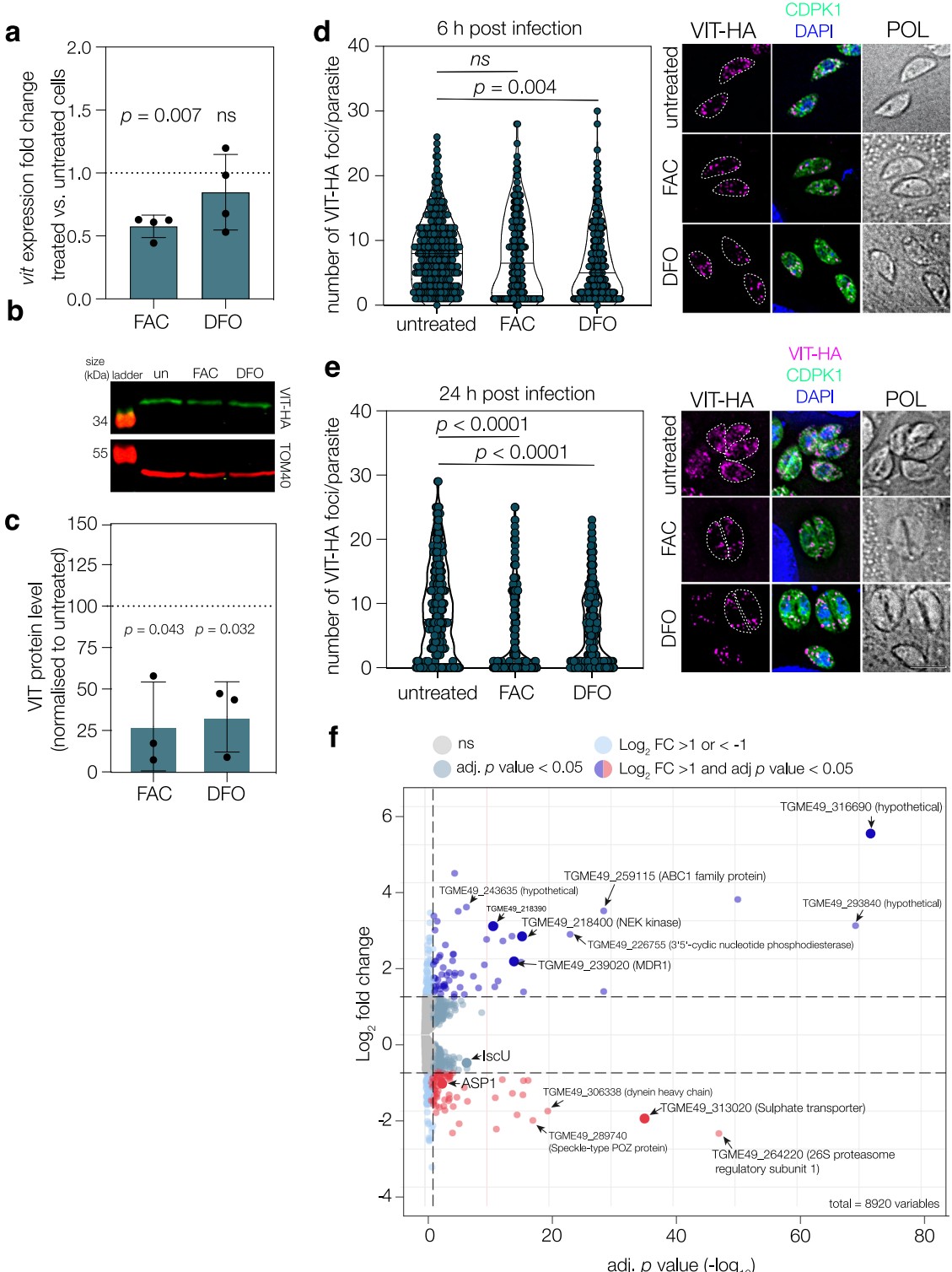

**Fig. 4 | VIT expression is regulated by the changes in iron levels. a** qRT PCR on *vit* transcripts after treatment with FAC or DFO, normalised to actin. Points represent 4 independent experiments, bars at mean ± SD. *p* values from two tailed one sample *t*-test. **b** Western blot showing levels of VIT-HA after 24 h treatment with 100 μM DFO or 5 mM FAC. TOM40 used as a loading control. **c** Quantification of VIT-HA levels from three independent experiments, bar at mean ± SD, *p* values from one sample two tailed *t* test. **d** VIT-HA foci at 6 h post invasion after treatment with FAC or DFO. There was no significant change upon FAC treatment, however, there was a significant decrease in the number of foci upon DFO treatment. POL polarised light.

Results from 3 independent experiments, *n* = 300 parasites. *p* values from one way ANOVA with Tukey correction. **e** As above, but quantified at 24 h post invasion. There was a significant decrease in the number of foci after treatment with both FAC and DFO, *p* values from one way ANOVA with Tukey correction. Results from 3 independent experiments, *n* = 300 parasites. *p* values from one way ANOVA with Tukey correction. **f** Volcano plot from RNAseq data comparing the parental line to ΔVIT. Adjusted *p* values from Wald test with Benjamini and Hochberg correction. See text for more details.

increase in ROS levels under normal conditions in the ΔVIT strain compared to the parental. However, upon FAC treatment ROS levels were significantly ($p = 0.024$, one way ANOVA with Sidak's correction) higher in the ΔVIT strain than in the parental and significantly raised ($p = 0.003$, one way ANOVA with Sidak's correction) compared to the untreated ΔVIT line (Fig. 5a).

To determine if the increase in ROS induced by FAC in the ΔVIT line is responsible for the increased sensitivity, we treated the parasites with the ROS scavenger N-acetyl-cysteine (NAC)[50]. Upon 5 mM NAC treatment, there was a significant ($p < 0.0001$, extra sum of square F test) protective effect ($EC_{50}$ 3.8 μM, 95% CI: 0.08 to 5.5 μM, compared to 0.018 μM, 95% CI: 0.0021 to 0.1 μM) to excess iron in the ΔVIT parasites (Fig. 5b, c), while the $EC_{50}$ for the parental line increased slightly but not significantly (from 88.28 μM, 95% CI: 24.4 to 307 μM to 109.7 μM, 95% CI: 15 to 136 μM) (Fig. 5c). This demonstrates that a significant driver of the increased sensitivity of the ΔVIT parasites to FAC is the increased production of ROS.

Although the ΔVIT parasites have a small growth defect in vitro, we were intrigued to see that there was no increase in ROS under normal conditions. *T. gondii* tachyzoites have various lines of defence against ROS, including thioredoxins, two SODs and a cytosolic catalase[6,51,52]. The RNAseq did not show any changes in SOD, SOD2 (Fig. S8a) or catalase transcription (Fig. S8b). We investigated the activity of these enzymes. SOD activity was visualised using a native page in-gel activity assay[53], however, we saw no obvious difference between the parental and ΔVIT parasites (Fig. S8c). *T. gondii* encodes three SOD enzymes, two of which are expressed in tachyzoites and localised to the cytosol and mitochondria[6,53] we were only able to distinguish one from the host cell SOD enzymes. Given this, we cannot exclude the possibility that SOD activity overall did change, but we were unable to assess it in this assay. We also quantified the enzyme activity of catalase using a colorimetric assay[54,55]. We found that deletion of VIT led to a significantly ($p = 0.0009$, Welch's one-way ANOVA with Dunnett's correction) increased catalase activity, compared to the parental line under normal growth conditions (Fig. 5d). Interestingly, we saw an apparent decrease in catalase activity (although this was not significant) in the ΔVIT line upon FAC treatment. It has previously been reported that oxidative stress can decrease catalase activity in plants[56] and yeast[57] and could explain why we only see this inhibition of activity in the ΔVIT strain, which is under the highest levels of oxidative stress (Fig. 5a). However, under normal conditions, the increase in catalase activity suggests that to maintain growth, ΔVIT parasites require higher catalase activity. This is despite the lower total amounts of parasite-associated iron (Fig. 2h), potentially due to the mislocalisation of iron in these parasites.

In *T. gondii*, catalase is cytosolic[6] and would not protect the parasite's mitochondrion from ROS. To examine if excess iron alters mitochondrial ROS (mROS) accumulation, we stained parasites with the mitochondrial-specific ROS probe MitoSOX (Fig. 5e, f) after confirming that MitoSOX localised to the parasite's mitochondrion (as defined by MitoTracker) in live cells (Fig. S8d). Quantifying MitoSOX fluorescence by FACS, we found that excess iron led to an increase in mROS in the parental line, however, this was significantly exacerbated in ΔVIT parasites. Interestingly we also saw a significantly higher percentage of MitoSOX-high cells in the ΔVIT under normal growth conditions, in contrast to the cytosolic ROS results (Fig. 5a). Despite the higher levels of mROS, we did not see a significant change in mitochondrial membrane potential as measured by MitoTracker (Fig. S8e). Upon examination of the literature, we found that overexpression of a mitochondrial iron transporter can compensate somewhat for the lack of CCC1 in yeast[58]. Through reciprocal BLAST searches, we found the parasite homologue of the mitochondrial iron transporter, TGME49_277090 (Fig. S9a). Based on homology and in common with mammals, we called this putative transporter mitochondrial iron transporter (MIT) and endogenously tagged this gene in the parental

and ΔVIT parasite lines (Fig. S9b). MIT-HA localised to the mitochondrion, as demonstrated by co-localisation with the mitochondrial marker TOM40 (Fig. 5g)[59]. Interestingly, we saw an increase in MIT-HA fluorescence in the mitochondrion in the ΔVIT line. We quantified this from immunofluorescence images and saw an increase in MIT-HA staining in the mitochondrion of the ΔVIT line compared to TOM40 (as an internal control) (Fig. 5h). An increase in total protein was confirmed by western blotting of MIT-HA (Fig. 5i, j). These results suggest that in the ΔVIT line, parasites upregulate MIT at the protein level, perhaps in an attempt to remove iron from the cytosol.

## VIT contributes to virulence in vivo

The increased activity of catalase and the reduced growth in cells suggested that VIT has an important role in survival of *T. gondii* in vitro. To determine if VIT had a role in pathogenesis, we infected ten Swiss Webster mice peritoneally with either 20 or 100 tachyzoites of the parental or ΔVIT::DHFR$_{TS}$ parasites and assessed mouse survival over time (Fig. 6a). Deletion of VIT led to significantly increased survival after infection with both 20 ($p < 0.0001$, log-rank test) and 100 parasites ($p = 0.0027$, log-rank test) demonstrating that VIT has an important role in virulence in vivo.

Given the relatively small growth defect seen under normal growth conditions, we were surprised by the magnitude of the effect in vivo. In order to cause disease, *T. gondii* must evade the host immune system and colonise various tissues. One important bottleneck is the ability of the parasite to survive in activated phagocytic cells[60,61]. To determine if loss of VIT affected the parasite's ability to survive in highly stressful environments, we prestimulated the murine monocytic cell line RAW 264.3 with LPS and interferon-γ (IFNγ-) before infecting them at an MOI of 5 for 48 h. We quantified parasite fluorescence and normalised levels in stimulated macrophages to unstimulated cells. ΔVIT parasites demonstrated less fluorescence even in unstimulated macrophages (Fig. 6b), possibly related to the reduced ability of these parasites to replicate (e.g., Fig. 1f). However, after normalisation we saw that ΔVIT parasites had a significantly ($p = 0.01$, *t* test) reduced ability to survive in activated macrophages, with a survival of around 12% compared to 21% survival in the mNeon line (Fig. 6c). Although virulence is a complex process, previous studies have identified genes required for survival in activated macrophages which have a defect in virulence[61], supporting our hypothesis that the reduced survival of the ΔVIT parasites in macrophages contributed to their inability to cause severe disease in mice.

## Discussion

Iron plays a central role in metabolism; however, its potential toxicity means that transport and storage are tightly regulated. Cells lacking the ability to correctly detoxify and store iron typically develop iron hypersensitivity, excess ROS and are frequently unable to survive other stresses[12,23,62–64]. Here, we have investigated the role of VIT in iron storage and detoxification in *T. gondii*. Deletion of VIT leads to a decrease in stored iron within the cells combined with iron hypersensitivity. We find that VIT localises to a dynamic compartment within the cell, which changes in distribution throughout the lytic cycle. VIT is regulated at the transcript and protein level by iron availability and the absence of VIT-mediated iron detoxification leads to transcriptional changes in the cell, including in pathways and genes known to rely on iron. We show that in the absence of VIT, parasite hypersensitivity to excess iron can be reversed by scavenging of excess ROS, suggesting that iron-generated ROS is the major cause of parasite death and that parasites lacking VIT can compensate in multiple ways, by raising antioxidant defences and possibly moving iron into alternative organelles. Finally, we show that VIT is required for survival in activated macrophages, and in pathogenesis in vivo, confirming the importance of iron detoxification to the parasite.

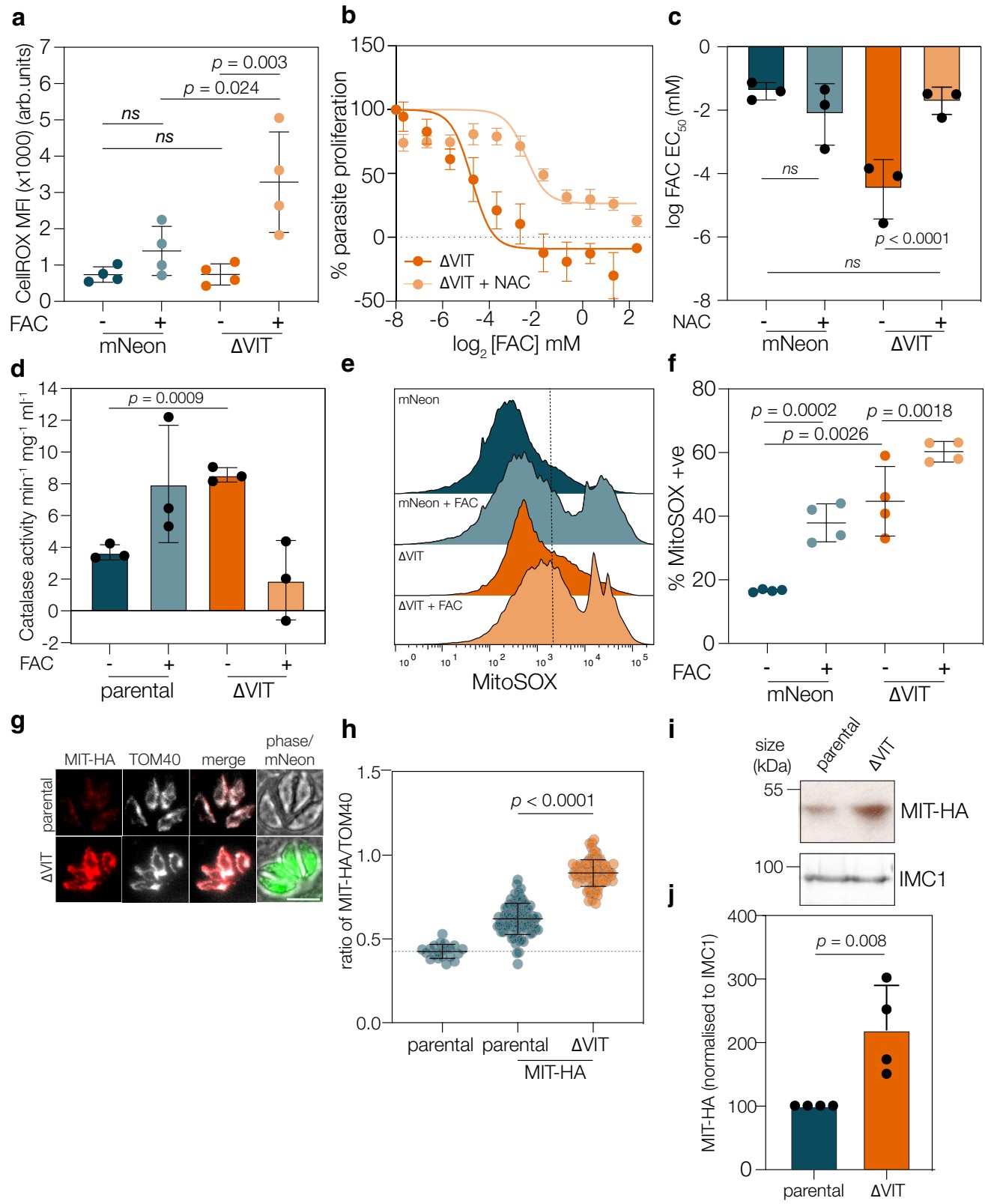

Lack of VIT causes hypersensitivity to excess iron in *T. gondii* which could be partially complemented through expression of a further copy under the endogenous promoter. The reason for this partial complementation is not known, however, overexpression of VIT from the tubulin promoter appeared toxic (results not shown) and in the complemented line *vit* levels were approximately double that of the parental strain. It is possible that tight control of *vit* levels, through both the promoter and 3' UTR are required for optimal response to excess iron. The dramatic iron hypersensitivity phenotype is mostly specific to iron, as no change in the sensitivity to other metals was seen (apart from the highly toxic cadmium). This supports previous data from *Plasmodium* showing that apicomplexan VIT is a selective iron transporter[16], unlike the homologues from rice or yeast which additionally transport zinc and manganese[12,23]. Further supporting VIT specificity, we saw no changes in

**Fig. 5 | Iron overload in absence of VIT leads to ROS accumulation. a** mNeon and ΔVIT parasites were treated with FAC and CellROX Deep Red fluorescence quantified using flow cytometry. FAC-treated parental cells had no significant effect on CellROX fluorescence, however, in ΔVIT parasites, CellROX was significantly ($p = 0.004$, one way ANOVA with Sidak correction) increased. Each point represents geometric mean fluorescence of over 10,000 cells from $n = 4$, ns not significant. Bars are at mean, ±SD. **b** Treatment with NAC (5 mM) rescued ΔVIT parasite hypersensitivity to FAC. Points are the mean of $n = 3$, ±SEM. **c** Graph showing the mean EC$_{50}$ for FAC of mNeon and ΔVIT parasites, with or without NAC treatment. Each point represents an individual experiment, bars at the mean of $n = 3$, ±SD. $p$ values from two sided extra sum of square $F$ test. **d** ΔVIT parasites had significantly higher catalase activity compared to the parental line. Each point represents an independent experiment, $n = 3$, bar at mean ± SD. $p$ value from one way ANOVA

with Sidak correction. **e** mNeon and ΔVIT parasites were treated as in (**a**), and MitoSOX fluorescence quantified. Overlapping histogram from a single experiment showing change in fluorescence distribution. **f** Percentage of cells MitoSOX positive from four independent experiments, bars at the mean ± SD. $p$ value from one way ANOVA with Tukey correction. **g** Immunofluorescence showing localisation and expression of parental and ΔVIT parasites endogenously expressing MIT-HA. Scale bar 5 um. **h** Quantification of MIT-HA signal as a ratio of TOM40. Points represent individual vacuoles from $n = 3$, bar at mean ± SD. $p$ value from one way ANOVA with Tukey correction. **i** Western blot confirming the increased expression of MIT-HA in the ΔVIT parasites compared to the parental line, IMC1 used as loading control. Representative of 4 independent experiments. **j** Quantification of MIT-HA expression, normalised to loading control. Bar at mean ± SD, $n = 3$ individual biological replicates. $p$ value from two tailed one sample $t$ test.

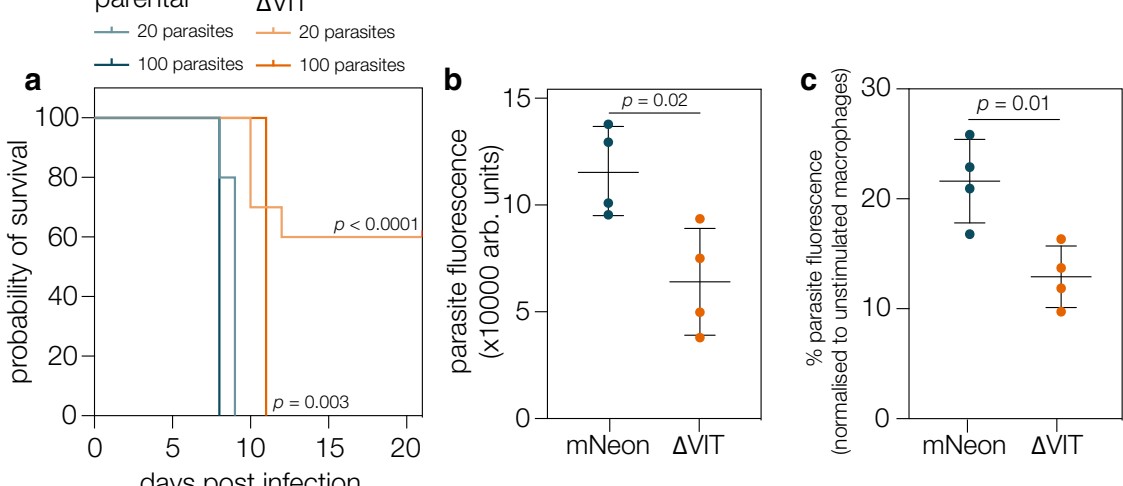

**Fig. 6 | VIT contributes to parasite survival in macrophages and for pathogenesis in vivo. a** 10 mice were infected with 20 or 100 tachyzoites of the indicated strain and survival monitored over the course of the experiment. $p$ values from Logrank (Mantel−Cox) test. **b** mNeon and ΔVIT parasites were used to infect macrophages at an MOI of 5 and fluorescence (arbitrary units) monitored after 72 h. Results are the mean of $n = 4$ independent biological replicates, performed in

triplicate, ±SD. $p$ values from two tailed $t$ test. **c** After normalisation of the fluorescence in the unstimulated macrophages (from **b**) there was a significant decrease in survival of the ΔVIT parasites in stimulated (IFNγ/LPS) macrophages. Results are the mean of $n = 4$ independent biological replicates, performed in triplicate, ±SD. $p$ values from two tailed $t$ test.

parasite-associated zinc upon VIT deletion and using XFM we saw that zinc and iron appeared to localise in separate compartments within the cells. In *T. gondii*, lack of VIT was also associated with decreased parasite-associated iron and a decreased ability to replicate in acute iron deprivation. These results are congruent with studies on VIT transporters of yeast and plants which also showed that the absence of CCC1/VIT was associated with reduced cellular iron[12,65]. However, in *Plasmodium*, deletion of VIT was associated with an increase in chelatable, or labile, iron in the cells[16]. We were not able to assess the labile iron in our cells using a similar method. However, our results are likely consistent with data from *Plasmodium*, as although the total parasite-associated iron is reduced, we believe that the labile iron in the cytoplasm increases in the absence of VIT. Supporting this, we also see increased cytosolic ROS upon iron treatment compared to the parental line.

VIT had a highly dynamic localisation within the cell, existing as a single point in extracellular parasites and shortly after invasion before fragmenting as parasite replication was initiated. We saw limited co-localisation between VIT and cathepsin protease L (CPL) and VP1, markers of the plant-like vacuolar compartment (PLVAC)[34–36]. The PLVAC is a dynamic, membrane-bound vesicular structure which appears to act as a site for ingested protein digestion[66], ion storage[18,35] and protein trafficking[67]. Localisation of VIT to the PLVAC fits well with previous observations that VIT is an iron/H$^+$ antiporter[14,20] as the PLVAC is acidified[34–36].

In contrast to our results, in *P. berghei*, VIT-GFP was found to co-localise with markers for the ER throughout the parasite's lifecycle[16]. It is possible that this differential localisation was influenced by the tags used (e.g., the presence of GFP can alter the localisation of proteins[68,69]. However, the differences in the localisation observed between *T. gondii* and *P. berghei* may well also reflect fundamental differences in the biology of these parasites. *Plasmodium* parasites replicate asexually within red blood cells which contain large amounts of iron-bound haemoglobin. For this reason, in *Plasmodium* the vast majority of parasite-associated iron is sequestered as hemozoin crystals within the digestive vacuole[19,70]. In contrast, *T. gondii* lives in the relatively iron-poor environment of nucleated cells. This difference in environmental conditions may be driving the differences in localisation observed. Further, ER-resident *ccc1*-homologues have been reported in plants[71], suggesting that there is a potential role for the ER in iron storage and detoxification. Despite potential differences in localisation, the phenotype of VIT in both species suggests a surprising overlap in mechanisms of iron detoxification. Further studies, examining the localisation of VIT in greater detail in *T. gondii* and *Plasmodium*, may be helpful to understand the location and dynamics of iron storage in both species.

Our data reveal that VIT levels drop transcriptionally upon iron excess, and at the protein level both upon iron depletion and excess. Transcriptional and post-transcriptional control of iron uptake and

storage have been examined in detail in bacteria, yeast, plants and mammalian cells[40,42,72–74], and was recently also demonstrated in the kinetoplastid parasite *Trypanosoma brucei*[75]. However, iron-mediated regulation has not previously been observed in the Apicomplexa. Interestingly, upon excess iron we saw a drop in transcript and protein levels of VIT, the opposite response to the predicted upregulation as seen in yeast[12]. The mechanism for regulation of VIT remains unknown, it is possible that while transcription and translation are dampened, the localisation or activity of the protein is upregulated, allowing the cells to adapt to higher iron levels. However, this remains to be further investigated. Recently, expression of an amino acid transporter, ApiAT1, has been shown to be regulated by substrate availability via an upstream small ORF in *T. gondii*[41]. VIT now joins ApiAT1 as the only substrate-regulated transporters identified in the Apicomplexa, although our results suggest that VIT may be regulated at multiple levels, including by modulating localisation. Understanding when and how each method of regulation is used by the parasite, and the pathways driving this control, remains to be unpicked.

Obtaining a clean knockout of VIT allowed us to assess how the parasites compensated for the lack of iron storage. It has been shown that in bacteria, yeast, and some plants under stressful conditions, cells will induce iron sparing programmes, reducing the production of some iron-containing proteins while prioritising others, often orchestrated at the transcriptional level[73,76–78]. Our findings provide the first evidence of such programmes in *T. gondii*. In the absence of VIT, *T. gondii* represses IscU while upregulating IscA. Transcriptional control of IscU by iron levels has been observed in a number of organisms[47,48] and regulation of an IscA homologue by iron has been observed in bacteria[72]. Given the recent interest in iron-sulfur biogenesis in *T. gondii* as a target for drug development[5,79] these findings suggest a currently unknown mechanism regulating this essential metabolic pathway. Beyond transcription, we found that under normal conditions, ΔVIT parasites attempted to compensate by increasing the activity (although not expression) of catalase and possibly other antioxidants, however this increase was not sufficient to prevent ROS accumulation upon further stresses. The importance of iron storage and detoxification across the tree of life[49,80] highlights the dangers of iron to almost all cells. We also saw an increased expression of a putative mitochondrial iron transporter, MIT in the absence of VIT. We suggest that in the absence of correct iron storage, *T. gondii* upregulates this transporter, increasing the iron (and ROS) levels in the mitochondrion, to help remove iron from the cytosol. A similar pathway has been proposed in yeast[58] and recent work in human cells demonstrated that overexpression of mitochondrial iron transporters was sufficient to overcome iron depletion[81]. Our results suggest a level of flexibility in iron compartmentalisation in *T. gondii* in in vitro growth conditions.

However, VIT appears to play a key role in harsher environments. Activation of macrophages by IFNγ leads to not only induction of pathogen killing programmes, but also nutrient restriction, including reduced levels of iron[82–84]. Further, a previous study on IFNγ-activated enterocytes saw that *T. gondii* growth restriction was reversible by addition of excess iron[85]. Given that the ΔVIT parasites are more susceptible to acute iron deprivation, this immune-stimulated nutrient restriction may be enough to depress replication, although we cannot rule out that the parasites are more susceptible to the stressful conditions of the phagosome. However, this defect in survival under immune pressure may lead to the decreased virulence seen in vivo.

In conclusion, we provide the first study of a proposed iron transporter in *T. gondii*. We show that it has a role, although is not essential, for growth under normal conditions, but is vital in detoxification of excess iron. This work provides a basis for the future study of iron within *T. gondii* where many questions remain, including how iron enters the parasite, how it moves between the organelles and the mechanisms of iron-mediated regulation. The conservation of VIT between the Apicomplexa demonstrates that, despite the very different environments that the parasites make home, they utilise common mechanisms to limit environmental toxicity.

## Methods

Animal studies described here adhere to a protocol approved by the Committee on the Use and Care of Animals of the University of Michigan.

### *T. gondii* and host cell maintenance

*Toxoplasma gondii* tachyzoites were maintained at 37 °C with 5% $CO_2$ and were grown in human foreskin fibroblasts (HFFs) cultured in Dulbecco's modified Eagle's medium (DMEM) that was supplemented with either 3% (D3) or 10% heat-inactivated foetal bovine serum (D10) (FBS), 2 mM L-glutamine and 10 µg ml$^{-1}$ gentamicin.

### *T. gondii* strain construction

The ΔKu80::mNeon and ΔKu80::tdTomato strains were generated by transfection of 40 µg of plasmids containing Cas9 with the required sgRNA (sgRNA sequences are listed in Table 1) along with 30 µg of purified PCR product (mNeon green expressing cassette) - amplified using primers p1 and p2 (see Table 2 for primer sequences) into the RHΔHX line for the ΔKu80::mNeon line and with 30 µg of purified PCR product (tdTomato red expressing cassette) - amplified using primers p3 and p4 for the ΔKu80::tdTomato line. Parasites were transfected into the parental RHΔHX line using a Gene Pulse BioRad electroporator as previously described[86]. Fluorescent parasites were isolated by FACS using an Aria III (BD Biosciences), sorted directly into a 96 well plate pre-seeded with HFFs and incubated at 37 °C with 5% $CO_2$ for 5–7 days. Positive fluorescent plaques were then identified using microscopy. Absence of *ku80* was confirmed by PCR using primers p7 and p8.

The ΔVIT strain was constructed by co-transfection of two Cas9 guides targeted to the 5′ and 3′ ends of the genomic sequence with a repair template consisting of an mNeonGreen cassette (including a strong SAG1 promoter) in the opposite orientation from *vit*. Parasites were sorted and cloned out as above and integration of the repair cassette confirmed by PCR using primers 20 and 21 for the 3′ junction, primers 22 and 23 for the 5′ junction and 24 and 25 to amplify the endogenous gene. The ΔVIT::DHFR$_{TS}$[87] was constructed by co-transfection of two Cas9 guide plasmids harbouring a fluorescently tagged Cas9 and a Bleomycin resistance cassette targeted to the 5′ and 3′ ends of the genomic sequence (Table 1) with a repair template consisting of the coding region of the pyrimethamine resistance cassette using primers p13 and p14. Transgenic parasites were selected with pyrimethamine, cloned by limiting dilution, and confirmed by PCR using primers p19 and p24.

The VIT-HA strain was constructed by transfection of a Cas9 guide targeting the 3′ end of the genomic sequence (Table 1) and repaired with an annealed synthetic oligonucleotide (IDT) encoding 40 base pairs of homology and inserting three copies of the HA epitope sequence prior to the stop codon. Fifty million tachyzoites were transfected by electroporation in a 4 mm gap cuvette using a Bio-Rad Gene Pulser II with an exponential decay programme set to 1500 V, 25 µF capacitance and no resistance and immediately added to a

**Table 1 | gRNA sequences used in this study**

| Name | Sequence (5′–3′) |
| --- | --- |
| *ku80* 5′ | cacttgggacggcgttgaat |
| *ku80* 3′ | cgcggccgtcatcgcagttt |
| *vit* 5′ | gtccctttttccacgaatt |
| *vit* 3′ | aacttgactgccgcctagag |
| *mit* 3′ | gtgaacgctctcccgatttg |
| *uprt* | ggcgtctcgattgtgagagc |

## Table 2 | Primer sequences

| Name | Sequence (5'–3') | Notes |
|---|---|---|
| p1 | aggatgcgccaccccgcctgccaacccgagcttggcttgcAAGCTTTACATCCGTTGCC | *mNeon for vit KO FW* |
| p2 | ccgaaagcttgaccgtgtgtcgacagacgtcaacaagaCCCtcgggggcaagaatt | *mNeon for vit KO RV* |
| p3 | CGCGTCTTCTGGAGGTGCCTCTTCGCCGTCTCCTCTTTCCTCGAGGAAGCTTTTACATCC | *mNeon for ku80 KO FW* |
| p4 | GATCGAGTAAGTCATCAACGTCATGCGGATCTAAGGGTCTCCCTCGGGGGGCAAGAATT | *mNeon for ku80 KO RV* |
| p5 | CGCGTCTTCTGGAGGTGCCTCTTCGCCGTCTCCTCTTTCCCATGCATGTCCCGCGTTCGT | *tdTomato for ku80 KO FW* |
| p6 | GATCGAGTAAGTCATCAACGTCATGCGGATCTAAGGGTCTGTGTCATGTAGCCTGCCAGA | *tdTomato for ku80 KO RV* |
| p7 | GGCGCTTCCTGGCAAAGCTTCAAG | *ku80 FW* |
| p8 | CCGTCAATGGCGTCACTCCGATTCG | *ku80 RV* |
| p9 | CGCCCAGTCCATGGTTGATA | *VIT RTqPCR FW* |
| p10 | TGAACATGACCAAGCCTCTCT | *VIT RTqPCR RV* |
| p11 | GGGGACGACATGGAGAAAATC | *Actin RTqPCR FW* |
| p12 | AGAAAGAACGGCCTGGATAG | *Actin RTqPCR RV* |
| p13 | CAGACGTCAACAAGATCCAAATTCGTGGAAAAAAGGGACAATGCAGAAAACCGGTGTGTCT | *DHFR$_{TS}$ for vit ko FW* |
| p14 | GCCACCCCGCCTGCCAACCCGAGCTTGGCTTGCGTCGGTGCAAACTTGACTGCCGCCATCTCCATCT | *DHFR$_{TS}$ for vit ko FW* |
| p15 | ATGCCATGGTGTCCTCGCAGCTGTCGTCGGTGCAAACTTGACTGCCGCCATTAAAATTGGAAGTGGAGGACGG | *forward primer for the epitope-tagging of VIT with 3xmyc epitope at its C-terminus* |
| p16 | CCCAGGATGCGCCACCCCGCCTGCCAACCCGAGCTTGGCTTGCAGCCACTGCTCTAGAACTAGTGGATC | *reverse primer for the epitope-tagging of VIT with 3xmyc epitope at its C-terminus* |
| p17 | GTGCTCGCAGCTCGTCGTCGGTGCAAACTTGACTGCGCCGCCTACCCATACGATGTTCCA-GATTACGCTGGCTATCCCTATGACGTCCCGGACTATGCAGGATCCTATCCATATGACGTTCCAGATTACGCTTA-GAGTGGCTGCAAGCCAAGCTCGGGTTGGCAGGCGGGTGG | *Repair template for generation of VIT-HA FW* |
| p18 | CCACCCCGCCTGCCAACCCGAGCTTGGCTTGCAGCCACTCTAAGCGTAATCTGGAACGTCATATGGATAGGATCCTGCA-TAGTCCGGACGTCATAGGGATAGCCAGCGTAATCTGAACATCGTATGGGTAGGCGGCCAGTCAAGTTTGCACCGACGACGAGCGTGCGAGCAC | *Repair template for generation of VIT-HA RV* |
| p19 | GCAATCTCCCCCACCGAGAGGTC | *Primer from DHFR cassette for integration FW* |
| p20 | gggttgtctttcgtaccccgttgc | *Amplification of the 3' vit junction FW* |
| p21 | ctttataatgggcacatgc | *Amplification of the 3' vit junction RV* |
| p22 | ggacttcaggtttaactoctcataac | *Amplification of the 5' vit junction FW* |
| p23 | gccgacctccatgcatttgatatc | *Amplification of the 5' vit junction RV* |
| p24 | GTTCTTAGAAGGTCTCTATGCCATGG | *Confirmation of presence of vit gene FW* |
| p25 | gttcctctgaaacgttcaacgttgcagatggaatatgTACCCGTACGACGTCCCGGA | *Repair template for generation of MIT-HA FW* |
| p26 | tcgccacactgtctacagagagctgactagaaagttacacCTGATAGCTGTCTTGTGTAT | *Repair template for generation of MIT-HA RV* |
| p27 | ggtctcgctacctgcgtttctgcg | *Amplification of the 3' MIT junction RV* |
| p28 | GGATCCGAGCACGCGCAGTAAAG | *Amplification of the 5' MIT junction RV* |
| p29 | CACACATTCAAAAGAAAGAAAGAAAAAATATACCCCAGCCTCGAGATGCCAGCTAGCGGAGC | *Amplification of vit for yeast expression FW* |
| p30 | CAAAAGAAAGAAAAAAAATATACCCCAGCCTCGAgtgACCAACACCTCGTCGGACTACG | *Amplification of vit$_{65-306}$ (sVIT) for yeast expression FW* |
| p31 | GACCAAACCTCTGGGCAAGAAGTCCAAAGCTGGATCCTTAATTAAAATCGAGCGGGTCC | *Amplification of viT for yeast expression RV* |
| p32 | TACCATGGAGTTTCCTTCATTCCAAGATCTGTGGCGTCGATTGTGAGGGGTACCaacagcacgtgaaacggc | *Amplification of pvit-vit for insertion into uprt locus FW* |
| p33 | GGGCAGCCGCCGGCAAACTGCCCCGCCAAGCCGCGCTTTCCATCGACTCGACCGCTAAAAAATGTGGACACAGTCGG | *Amplification of pvit-vit for insertion into uprt locus RV* |

confluent HFF monolayer in a T25 flask. Parasites were selected with 50 µg/mL phleomycin at 24 h post transfection for 6 h (to select for presence of the sgRNA plasmid) and subsequently added to a confluent HFF monolayer in T25 flask. Parasites were cloned by serial dilution and insertion of the HA tag was confirmed by IFA and western blot.

The VIT-Myc strain was constructed by transfection of a Cas9 guide targeting the 3' end of the genomic sequence. The 50 bp upstream and downstream homologous regions around the stop codon of the VIT gene were encoded in the forward and reverse primers to generate the repair template by PCR (using premier p15 and p16, Table 2), which flanks the homologous regions at both ends of the 3xMyc epitope tag and chloramphenicol resistance cassette. The plasmid encoding the *TgVIT*-targeting guide RNA and Cas9-GFP and the repair template were co-transfected into RHΔ*ku80* parasites. The stop codon of *vit* was replaced by the 3xMyc epitope tag and chloramphenicol resistance cassette. The transfectants were selected through multiple rounds of chloramphenicol to generate a stable population. The VIT-Myc fusion protein was confirmed by immunoblotting analysis.

The MIT-HA strain was constructed by transfection of a Cas9 guide targeting the 3' end of the genomic sequence (Table 1). The 50 bp upstream and downstream homologous regions around the stop codon of the *mit* gene (TGME49_277090) were encoded in the forward and reverse primers to generate the repair template by PCR (using primer p25 and p26, Table 2), which flanks the homologous regions at both ends of the 3xHA epitope tag and chloramphenicol resistance cassette. Parasites were transfected as above and selected using chloramphenicol prior to cloning and the insertion of the HA tag verified by PCR using primers p27 and p28 and immunofluorescence.

The ΔVIT + VIT complemented strain was constructed by transfection of a Cas9 guide targeting the coding region of the *uprt* gene (Table 1). The 50 bp of upstream and downstream of the cut site of the *uprt* gene were encoded by the forward and reverse primers (p32 and 33, Table 2) to generate a repair template containing the endogenous *vit* promoter, *vit* cDNA and the *dhfr* 3' UTR. Parasites were transfected as above and selected for integration by 5 days of treatment with 10 µM 5-fluorodeoxyribose (FUDR), prior to cloning. The presence of the *vit* cassette and disruption of the endogenous locus was confirmed by PCR.

## Yeast strain construction

VIT and sVIT (missing the first 64 amino acids) were cloned into the yeast expression vector pRD195 (Addgene, #36028) using primers p29 and p31 for full length VIT and p30 and p31 for sVIT. Wild type (BY4742) and Δ*ccc1* yeast (a kind gift from Janneke Balk, John Innes Centre)[21] were transformed as previously described[88] and positive transformants were selected on synthetic dropout medium lacking uracil (SD-Ura). Overnight cultures of selected colonies were grown in SD-Ura, then spotted onto 2% (w/v) SD-Ura agar or SD-Ura agar supplemented with 3 mM or 5 mM ferrous ammonium sulphate (FAS). Plates were photographed after 4 days and are representative of two independent experiments.

## Immunofluorescence assays

Indirect immunofluorescence assays (IFA) were performed on both infected HFF monolayers or naturally egressed extracellular tachyzoites. Intracellular *T. gondii* parasites that were grown on HFFs on coverslips were fixed with 4% paraformaldehyde (PFA) for 20 min at room temperature and then washed with 1X phosphate buffered saline (PBS). Extracellular parasites for immunofluorescence assays were incubated in buffer A (116 mM NaCl, 5.4 mM KCl, 0.8 mM $MgSO_4.7H_2O$, 50 mM Hepes and 5 mM glucose)[35] spun onto poly-L-lysine (Advanced BioMatrix) coated coverslips, fixed with 4% PFA for 20 min at room temperature and washed once in PBS. Cells were permeabilized and

blocked in PBS/0.2% Triton X-100/2% Bovine Serum Albumin (BSA) (PBS/Triton/BSA), for 30 min at room temperature or 4 °C overnight. The slides were then incubated in a wet chamber with primary antibodies (rat anti-HA, Merk (11867423001) 1:1000 and guinea pig anti-CDPK1, a kind gift of Dr Sebastian Lourido (Whitehead Institute)[89], 1:5000, rabbit anti-CPL, a kind gift from Dr. Vern Carruthers (University of Michigan)[66], 1:500, mouse anti-MYC (Life Technologies) 1:400, anti-TOM40, a kind gift from Dr Giel van Dooren (ANU), 1:1000 in the PBS/Triton/BSA mixture for 1 h at room temperature. After washing, slides were incubated in a wet chamber with the secondary antibodies (Alexa Fluor 488 or 594 goat anti-Rat, Invitrogen, 1:1000 and Alexa Fluor 594 goat anti-Guinea Pig, Alexa Fluor 488 or 594 goat anti-mouse Invitrogen, all 1:1000) in PBS/Triton/BSA for 1 h at room temperature in the dark. After further PBS washes, cells were then mounted onto slides with Fluoromount with DAPI (Southern Biotech). Micrograph images were obtained using a DeltaVision widefield (Applied Precision) or Leica DiM8 (Leica Microsystems) microscope and processed and deconvolved using SoftWoRx and FIJI software.

## X-ray Fluorescence Microscopy (XFM)

Samples for X-ray fluorescence microscopy were prepared following the protocol outlined by Finney and Jin[90]. Briefly, SiN windows (Norcada, Edmonton, Alberta, Canada) were attached to the bottom of a 35 mm tissue culture dish and HFFs were seeded overnight and infected with RHΔ*Ku80*Δ*HXG* (parental) parasites or ΔVIT::DHFR$_{TS}$ parasites at an MOI of 3. Parasites were allowed to invade for 1 h at 37 °C with 5% $CO_2$, fixed with 4% PFA for 20 min at RT, and washed with TRIS-Glucose buffer (260 mM glucose, 9 mM acetic acid and 10 mM Tris buffer, pH 7.4). Elemental mapping was performed at the Advanced Photon Source Synchrotron X-ray microprobe at Sector 2, beamline station 2-ID-E with a beam focused to 0.3 micron using Fresnel Zone Plates using hard X-rays of 10.5 keV, provided by Kohzu double-crystal monochromator. Fluorescence signal from the raster-scanned samples was collected with a 4-element Vortex-ME4 Silicon Drift Detector (Hitachi High-Tech Science America, Chatsworth, California, US) positioned at 90° to the incident beam. Per pixel elemental fitting and quantification were obtained using the thin-film reference sample AXO 1X (AXO DRESDEN, GmbH, Germany). Analysis was performed using MAPS software[91]. Images shown representative of 10 parasites.

## Plaque assays

500 parental or ΔVIT parasites were applied to confluent monolayers of HFF cells. Parasites were allowed to replicate for 7 days undisturbed before being washed with PBS, fixed with ice cold 70% ethanol and stained using crystal violet stain (12.5 g crystal violet in 125 ml ethanol, diluted in 500 ml 1% ammonium oxalate) for 10 min at room temperature before washing with distilled $H_2O$ and imagining.

## Extracellular survival

To assess extracellular survival, parasites were mechanically released from host cells, counted, and diluted to 1 parasite/µl in D3 and incubated at 37 °C for the indicated time. 100 µl of parasites were then placed onto two wells of a 12 well plate of confluent HFF cells and allowed to form plaques (as above) for 5–7 days. Plaques were counted and normalised to parasites not incubated extracellularly.

## Competition assay

ΔKu80::mNeon, ΔKu80::tdTomato and ΔVIT extracellular parasites were counted and 16ormalizin to $2 \times 10^7$ cells/ml in HBSS + 1% FBS. Equal numbers of ΔKu80::mNeon and ΔKu80::tdTomato parasites, and ΔKu80::tdTomato and ΔVIT parasites were co-inoculated in a 1:1 ratio into host cells with either DMEM alone or DMEM supplemented with 200 µM FAC (2 technical replicates per condition). The mixed populations were subsequently passaged every 2 or 3 days for about

10 days. At every passage, parasites were filtered and collected for analysis using a BD Celesta, data acquired using FACSDiva software (BD Biosciences). Data were processed using FlowJo v10 (BD Biosciences).

## Fluorescent growth assay

Growth assays of fluorescent cells were performed using mNeon-expressing parasites in black, clear bottomed 96 well plates, preseeded with HFFs. Host cells were infected with 1000 either ΔKu80::mNeon or ΔVIT tachyzoites per well and were allowed to invade for 2 h at 37 °C with 5% $CO_2$. After 2 h, the media was supplemented with either FAC, DFO, Ars, $ZnSO_4$, $CuSO_4$, $CdCl_2$ at appropriate concentrations and incubated at 37 °C with 5% $CO_2$ for 4 days. 5 mM N-acetyl-cysteine (NAC, Sigma A7250) was added to all wells where appropriate. Fluorescence (at 594 nm emission) for each well was recorded at 4 days using a PHERAstar FS microplate reader (BMG LabTech). All experiments performed in triplicate wells, at least three independent biological replicates. Uninfected wells served as blanks and fluorescence values were normalised to infected, untreated wells. Dose-response curves and EC50s were calculated using GraphPad Prism 9 software, using the log(drug) vs response model, with a constant Hill slope of −1. Where indicated, models were compared using the Extra sum of squares F test, comparing the LogEC50 value.

## ICP-MS

Total parasite-associated iron was quantified using ICP-MS. Confluent HFF monolayers in culture flasks were infected with either mock, parental, or ΔVIT::$DHFR_{TS}$ parasites and cultured for 30 h. Parasites were harvested by mechanical lysis and filtration through 0.3 μm filters and quantified on a hemocytometer. Parasites were washed with chelexed PBS and digested in 30% Nitric acid at 85 °C for 3 h. Digested samples were diluted in 2% Nitric acid with 0.025% Triton X-100. Data were collected in triplicate, validated by monitoring of $^{45}Sc$ as an internal standard with relative intensities within 60–125% of the response in the calibration blank. A four-point calibration curve was generated (5, 10, 25, and 50 ppb Fe) and $^{56}Fe$ was quantified. Syngistix version 2.2 was used for data collection and analysis.

## Alamar blue

HFF cells were grown in black opaque 96-well plates until confluent and were treated with indicated concentrations of sodium arsenite (Ars). This was then followed by an Alamar Blue assay with 0.5 mM resazurin at 37 °C with 5% $CO_2$ for 4 h. PHERAstar FS microplate reader (BMG LabTech) was used to determine fluorescence at 530/25 excitation and 590/25 emission. Host cell viability was determined by ormalizing values to untreated HFFs. The experiment was performed in triplicate for three biological replicates.

## RT-qPCR

RT-qPCR was used to assay relative *vit* expression. Human foreskin fibroblasts were incubated in either D3 or D3 supplemented with ferric ammonium citrate (5 mM) for 6 h prior to infection with *T. gondii* (RHΔKu80). Parasites were cultured for 24 h prior to collection by scraping the HFF monolayer, passing the cell suspension through a 26 G needle six times and filtration to remove host cell debris. Parasites were then pelleted by centrifugation at $1500 \times g$ for 10 min, the media was then removed and cell pellets frozen and stored at −80 °C. Total RNA was extracted from parasite pellets using the RNAeasy Mini kit (Qiagen) and DNAse I (Invitrogen) treated for 15 min at room temperature followed by DNAse denaturation at 65 °C for 20 min. cDNA synthesis was performed with the High Capacity cDNA RT kit (Invitrogen) according to manufacturer's instructions.

RT-qPCR was carried out on the Applied Biosystems 7500 Real Time PCR system using Power SYBRgreen PCR master mix (Invitrogen), 2 ng of cDNA per reaction and the following cycling conditions: 95 °C for 10 min, 40 cycles of 95 °C for 15 s, 60 °C for 1 min. Primer sequences can be found in Table 2. Reactions were run in triplicate from three independent biological experiments. Relative fold changes for treated vs. untreated cells was calculated using the $2^{-\Delta\Delta Ct}$ method[92] using actin as a housekeeping gene control. Graphpad Prism 9 was used to perform statistical analysis. Specifically, *t* tests testing the null hypothesis – that for each treatment $ddCt_{average} = 0$, were used to examine the statistical significance of the observed fold changes.

## Cellular thermal shift assay

Cellular thermal shift assays (CESTA) were performed as previously described[93] with some modifications. Briefly, approximately $5 \times 10^9$ VIT-HA expressing parasites were collected, split into two samples and incubated at 37 °C for 1 h in 800 μl Ringer's solution ((115 mM NaCl, 3 mM KCl, 2 mM $CaCl_2$, 1 mM $MgCl_2$, 3 mM $NaH_2PO_4$, 10 mM HEPES, 10 mM glucose)), (untreated) or Ringer's solution containing 2 mM ferric ammonium chloride (FAC). Cells were then divided into 9 PCR tubes and incubated at the following temperatures for 3 min, (37, 41, 43, 47, 50, 53, 56, 59, 63 °C). Parasites were moved onto ice, 16 μl of 6x lysis buffer (final concentration of 0.8% IGEPAL/NP-40, 1x Halt protease inhibitor (Sigma)) added and cells incubated for 10 min on ice. To ensure full lysis, cells were frozen on dry ice and thawed at RT three times. To remove protein aggregates, cells were spun at $14,000 \times g$ for 80 min at 4 °C, the supernatant was carefully removed and western blotted, as described below.

## Immunoelectron microscopy

For immunolabeling, the samples were fixed in phosphate buffer, pH 7.2, containing 4% freshly prepared formaldehyde. After several washes in the same buffer, they were dehydrated in ascending ethanol series and embedded in LR White resin (Agar Scientific). Ultrathin sections (70 nm thick) were obtained using an ultramicrotome (Leica Microsystems). The sections were collected on formvar-coated nickel grids and then blocked in PBS containing 3% bovine serum albumin for 1 h. After this time, they were incubated in the presence of anti-HA (Cell Signalling (C29F4)). Then they were washed several times in blocking buffer, and incubated with 15 nm gold-conjugated Protein A (Aurion). The grids were washed several times in the blocking buffer, dried and contrasted with 4% uranyl acetate, and observed using a JEOL 1200 EX transmission electron microscope operating at 80 kV.

## RNAseq

T75 flasks of human foreskin fibroblasts were incubated in either standard D3, D3 supplemented with 5 mM FAC or 100 μM DFO for 24 h prior to infection. Each T75 was then infected with $6–7 \times 10^6$ cells of wild type (RHΔKu80) or ΔVIT. Infected monolayers were cultured at 37 °C, 5% $CO_2$ for 24 h prior to parasite collection. Parasites were pelted by centrifugation at $1500 \times g$ for 10 min and stored at −80 °C as dry pellets until required. RNA was extracted from the pellets using the RNAeasy kit (Qiagen) according to the manufacturer's instructions. RNA libraries were then prepared using Illumina Stranded mRNA library preparation method and sequenced at $2 \times 75$ bp to an average of more than 5 million reads per sample. Raw sequencing data (FASTQ format) was processed using the Galaxy public server hosted by EuPathDB (https://veupathdb.globusgenomics.org/). FastQC and Trimmomatic were used for quality control and to remove low quality reads (where Q < 20 across 4 bp sliding windows) and adaptor sequences[94]. The filtered reads were aligned to the *T. gondii* ME49 genome using HITSAT2[95]. These sequence alignments were used to identify reads uniquely mapped to annotated genes using Htseq-count. Differential expression analysis was performed in R using DESeq2[96]. Additional packages EnhancedVolcano[97], pheatmap and ggplot2 were used to generate plots. All processed data is available in Supplementary Data 1 and raw FASTA files are available at http://www.ncbi.nlm.nih.gov/bioproject/, bioproject ID: PRJNA754376.

## Western blotting

Parasites were filtered and collected by centrifugation at $1500 \times g$ for 10 min. Samples were lysed with RIPA lysis buffer (150 mM sodium chloride, 1% Triton X-100, 0.5% sodium deoxycholate, 0.1% sodium dodecyl sulphate (SDS) and 50 mM Tris, pH 8.0) for 15 min on ice. Samples were then resuspended with 4X SDS loading dye with 5% w/v beta-mercaptoethanol, boiled at 95 °C for 5 min and separated on a 10% SDS-PAGE gel for 1.2 h at 160 v. EZ-Run Prestained protein (Fisher) ladder was used as a molecular weight marker. Proteins were then semi-dry transferred to nitrocellulose membrane in Towbin buffer (0.025 M Tris, 0.192 M Glycine, 10% methanol) for 30 min at 190 mA and blocked at room temperature in 5% milk in 0.1% Tween/PBS. Blots were then stained with the following antibodies for 1 h at room temperature: rat anti-HA (Merck) at 1:500 and rabbit anti-Tom40[59] at 1:2000 or guinea pig anti-CDPK1[89] at 1:10,000 followed by secondary fluorescent antibodies: goat anti-rat coupled to IRDye 800 (1:10,000, LI-COR) and goat anti-rabbit coupled to IRDye 680CW (1:10,000, LI-COR) and visualised using the Odyssey LCX. MIT-HA could not be imaged on the LI-COR system but was stained using rat anti-HA followed by anti-rat-HRP (1:5000, Abcam, ab6734)) and detected using Pierce ECL Western Blotting Substrate (Thermo Scientific).

## Flow cytometry

CellROX staining was used to quantify ROS as previously described[98]. Briefly, ΔKu80::mNeonGreen or ΔVIT parasites were untreated or treated with 2 mM FAC overnight, collected and washed once with PBS. Cells were stained with 10 μM CellROX Deep red (Thermo Fisher, C10422), 5 μM MitoSOX (Thermo Fisher, M36008) or 50 nM Mito-Tracker RedCMXRos (Thermo Fisher, M7512) or MitoTracker DeepRed (Thermo Fisher M22426) in PBS and incubated at 37 °C in the dark for 1 h (CellROX) or 10 min (MitoSOX and MitoTracker). Parasites, including unstrained controls, were then analysed on a BD Celesta analyser and data acquired using FACSDiva software (BD Biosciences). Parasites were gated on forward and side scatter and on green fluorescence. All data was analysed using FlowJo v10 (BD Biosciences). For CellROX, the geometric mean of CellROX fluorescence from at least 5000 parasites in the Cy5 channel was determined. For MitoSOX the geometric mean of the PE channel was used. Please see Fig. S10 for an indication of the gating strategy used.

## Catalase activity

Parental and ΔVIT parasites were untreated or treated overnight with 2 mM FAC and the activity of catalase assessed as described previously[54,55]. Briefly, parasites were collected, filtered, pelleted by centrifugation, and resuspended in 80 μl of molecular grade $H_2O$ with 20 μl of PBS. Parasites were lysed by freeze-thawing three times on dry ice before 50 μl of lysate was mixed with 100 μl of 10 mM $H_2O_2$ and incubated at 37 °C for 2 min. The lysate was then mixed with 600 μl of cobalt buffer (3.5 mM $Co(NO_3)_2 \cdot 6H_2O$, 3.2 mM sodium hexametaphosphate, $(NaPO_3)_6$ and 850 mM sodium bicarbonate) and incubated in the dark at room temperature for 10 min. Blank samples (only cobalt buffer) and standard (no lysate) controls were included. Absorbance was read at 440 nm using a PHERAstar plate reader and activity was calculated using Eq. (1):

$$\text{Catalase Activity of test kU} = 2.303(\text{time}) \times log\frac{\text{standard(absorbance)}}{\text{sample(absorbance)}} \quad (1)$$

## In gel SOD activity assay

Parasite SOD activity was assessed as previously described with some modifications[53]. Parental and ΔVIT parasites were scraped, syringed, filtered, and pelleted. Uninfected host cells were also scraped and syringed but the filtering step was omitted. Cell pellets were resuspended in 100 mM $KPO_4$ buffer supplemented with cOmplete, EDTA-free protease inhibitors (Sigma) and incubated on ice for 5 min. 10 μl was then reserved for western blotting while the remaining was mixed with native loading buffer (2 M sucrose, 0.5% bromophenol blue) and loaded onto a polyacrylamide gel in the abscess of SDS. The gel was ran at 150 v for 2–3 h at 4 °C before being washed gently in water and incubated in development solution (50 ml of 100 mM $KPO_4$, 50 mg Nitro Blue Tetrazolium, 7.5 mg riboflavin, 162 μl TEMED) for 45 min in the dark. The gel was then exposed to light and washed in distilled water 2–3 times before imaging.

## Phylogenetic analysis of MIT

Mitoferrin homologues were identified by reciprocal BLAST search against representative apicomplexan and model eukaryotic genomes. TGME49_235650 was used as an outgroup. Protein sequences were extracted, aligned using ClustalW and the phylogenetic tree generated by neighbour-joining and visualised using SeaView (v5.0.4)[99].

## Macrophage survival

$2.5 \times 10^6$/ml RAW 264.7 macrophages (ATCC TIB-71) were seeded onto a black, clear bottomed 96 well plate and allowed to adhere for 2–4 h. Macrophages were then activated by the addition of 0.2 μg/ml LPS and 0.1 ng/ml murine recombinant interferon-γ (IFNγ) (both ThermoFisher Scientific) for 24 h. Macrophages were infected with ΔKu80::mNeon or ΔVIT parasites at an MOI of 5 and fluorescence quantified at 48 h post infection using a PHERAstar plate reader. Survival was normalised to the fluorescence of naive macrophages. The experiment was performed in technical triplicate at least three times.

## In vivo infection

Groups of five female Swiss Webster (Jackson) female mice aged 6 weeks, were housed with a 12 h light-dark cycle at 20–24 °C with 30–70% relative humidity. Mice were infected with either 20 or 100 tachyzoites intraperitoneally in 150 μl of PBS. The experiment was performed twice and the results were combined. The number of injected parasites was confirmed by plaque assay. Surviving mice were tested for seropositivity by ELISA. Animal studies described here adhere to a protocol approved by the Committee on the Use and Care of Animals of the University of Michigan. Results were plotted in Graphpad Prism 9 and differences in survival assessed by the Log-rank (Mantel–Cox) test.

## Reporting summary

Further information on research design is available in the Nature Portfolio Reporting Summary linked to this article.

## Data availability

RNA seq data is available from http://www.ncbi.nlm.nih.gov/bioproject/ with the bioproject ID: PRJNA754376. Minimally processed data is present in supplementary dataset 1. Source data is provided with the manuscript. *T. gondii* genome information can be found at ToxoDB. Source data are provided with this paper.

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

## Acknowledgements

C.R.H. is funded by a Sir Henry Dale Fellowship from the Wellcome Trust and the Royal Society (213455/Z/18/Z) and a Carnegie Research Incentive Grant (RIG009880) from the Carnegie Trust. D.A. and C.R.H. are funded by a Lord Kelvin/Adam Smith (LKAS) Fellowship from the University of Glasgow. V.B.C. is funded by US National Institutes of Health grant R01AI120607. We acknowledge the University of Michigan Office of the Provost and the University of Michigan Departments of Chemistry and Biophysics for support to A.J.G. This research used resources of the Advanced Photon Source, a U.S. Department of Energy (DOE) Office of Science User Facility operated for the DOE Office of Science by Argonne National Laboratory under Contract No. DE-AC02-06CH11357. We thank Sebastian Lourido (Whitehead Institute, MIT) and Lilach Sheiner (University of Glasgow) for providing antibodies. The authors gratefully acknowledge the assistance of the Institute of Infection, Immunity and Inflammation Flow Core and Imaging facilities, especially Leandro Lemgruber, and Glasgow Polyomics for sequencing assistance. We thank Elizabeth Peat for invaluable technical assistance and Yara Aghabi for assistance in image quantification. We thank Tracey Schultz for invaluable assistance in performing the mouse infections.

## Author contributions

Conceptualisation: D.A., A.J.G. and C.R.H.; Resources: Z.D. and O.A.; Investigation: D.A., M.S., G.G., E.H., A.J.G. and C.R.H.; Writing–original draft: D.A., A.J.G. and C.R.H.; Writing–review and editing: D.A., M.S., A.J.G., V.B.C., Z. D. and C.R.H.; Funding acquisition: V.B.C., Z.D. and C.R.H.; Supervision, V.B.C., Z.D. and C.R.H.

## Competing interests

The authors declare no competing interests.
