## [Peer Review File · Nature Communications]

REVIEWER COMMENTS

Reviewer #1 (Remarks to the Author):

This manuscript investigates the role of a conserved organellar/vacuolar iron transporter VIT in storage/detoxification of iron in *Toxoplasma*. The rationale for this work is based on the fact that *Toxoplasma*, in contrast to its host cells, lacks the main iron storage protein ferritin. VIT was previously identified in plants and closely related *Plasmodium* and shown to be involved in transport/detoxification of iron. The structure of *Eucalyptus grandis* VIT has been solved. TgVIT was identified based on homology to its *Plasmodium* counterpart and shows significant conservation in residues important for iron binding. In this work, the authors generate various VIT null cells and show that these cells become hypersensitive to increased iron levels, although the nulls are viable. They show that increased sensitivity to exogenous iron in VIT nulls was due to production of excess ROS and this effect was reversed by addition of N-acetyl cysteine. Transient overexpression and epitope-tagging of VIT showed changes in its localisation from a single focus which became more dispersed through the lytic cycle. Colocalization of VIT with a vacuolar associated compartment marker suggests that this compartment substitutes for Fe storage systems that are used in other systems. Finally, VIT null cells have impaired survival in in vivo infectivity.

Iron-associated toxicity is an extremely interesting topic, not only for those interested in *Toxoplasma* biology, but also for higher eukaryotes – where many iron-associated disorders are lethal. While the experiments throughout the manuscript were done carefully with good quality data and rigorous statistical analyses, there are a few issues that I wish to highlight.

1. The evidence as presented for TgVIT conservation is weak. Fig 1b - The key amino acids shown does not reflect the full architecture and domain organisation of the VIT1 family across different species. Asp43 and Met80 involved in the coordination of metals is shown but other key residues involved in iron transport are not shown. From an evolutionary perspective it is important to show how many TM segments are present in TgVIT1, for example are Gly 44, 69, 76 in EgVIT1 (important for iron transport activity) conserved? In closely related PfVIT1, there are conserved glutamate residues in the cytoplasmic metal binding domain. Are these conserved in TgVIT1? I will suggest replacing the alignment with a schematic highlighting all key conserved residues: # of TMDs, conserved Gly, Glu and Met.

2. All the data presented rely on the fact that VIT nulls do not express VIT mRNA and thus protein. Based on the schematic in Fig 1c, the Knockout construct should result in replacement of all exons with mNeonGreen. In Fig 1d, PCR validation is presented with primers flanking regions within the 5' and 3'-ends of the repair template. If KO is achieved, Δ VIT1 cells should show no TgVIT1 transcripts, but this is not the case in Fig 4f, see RNA-Seq suppl Table S1. Relative to parental cells, VIT1 transcripts (geneID TGME49_266800) are only depleted by 60% ($\log_2FC = -0.7653217$, $padj = 0.00026238$) suggesting that

there is clear VIT transcription. This issue can be clarified by performing quantitative real time PCR of VIT transcripts in Parental vs Δ VIT cells and linking PCR using primers within the repair template and up/downstream flanks of the VIT1 locus – outside the repair template. The same holds true for all VIT null cells: mNeon (Fig 1), Δ ku80 mNeon, tdTomato, and DHFR. It is important to clarify which KO cell line was used for RNA-Seq and ensure that there are no VIT transcripts in all KO cells for all other analyses to be valid.

3. The conclusion that VIT localises to the vacuolar compartment may be correct, but it is compromised by three major issues. (i) transient overexpression (OE): I did not find much utility of these data as they don't reflect endogenous levels of VIT and VIT did not show increased expression in either low or excess iron. Secondly, the authors admit that VIT::Ty OE is toxic as these cells did not grow for prolonged periods. The conclusions from these data could be due to unavoidable excess protein that swamp the cells. The "detail" highlighted in Fig S4 is not convincing. I see no significant colocalization by eye. Most of the membrane localised outline seen in Ty is not present in HA. Myc and HA are convincing. (ii) in the EM in Fig 3i, some VIT in (i) is in the lumen of that vacuole. Isn't VIT a transmembrane protein? (iii) the authors argue that PfVIT may be mis-localised due to addition of GFP-tags. This explanation holds true for their data. One way to resolve this will be to knock-in HA- or Myc -epitope-tagged versions and show that these can rescue iron toxicity in VIT nulls.

4. While the title is not an overstatement, the evidence present is indirect. Direct and unequivocal evidence for intracellular iron detoxification could be provided by (i) showing that TgVIT binds iron, and (ii) fractionation of organellar vs cytosolic and measuring iron content in these compartments relative to total levels in null vs parental cells.

Minor points:

Fig 1a: Needs a scale bar to appreciate the localisation/dispersal of the different metals visualised across the panels shown. How many parasites were looked at per metal? Representative images of the number cells visualised should be stated.

Fig 1d. arrow indicates?

Fig 2a: Top panel should be Δ VIT not VIT, for consistency. I would have liked to see Parental and Δ VIT experiments at 50 and 200 micro molar not 10 and 500. Note: Δ VIT::DHFR cells are done at 50 and 200 micro molar. It makes comparisons consistent.

Fig 2b: 200 micro molar FAC does not affect survival of mNeon (2b) but what does the plague assay look like?

To what extent are the iron levels used in the in vitro experiments representative of physiological conditions. I found the choice of FAC concentrations at times excessive, 5 mM in RNA-Seq experiment and Western blot Fig 4 contrasting 200 μ M in Fig 2.

Fig S2b – It appears the gel image has been contrast-enhanced to mask a band in the parental. Please present original image

Reviewer #2 (Remarks to the Author):

In this manuscript, Aghabi and colleagues describe the role for a putative vacuolar iron transporter (VIT) in iron storage and homeostasis in *Toxoplasma gondii* parasites. The authors demonstrate that VIT has a critical role in mediating the parasite's response to iron detoxification, with loss of the transporter likely leading to increased production of reactive oxygen species in the parasite. They demonstrate a role for the transporter in mediating both parasite proliferation and virulence. They also present some evidence that the abundance of the VIT protein is regulated by exogenous iron levels, although the data on this are not entirely convincing (see comments below). Overall the manuscript breaks important new ground in understanding the responses of an intracellular parasite to the iron levels it encounters in its environment, and develops some powerful new approaches for analysing iron homeostasis in the parasite (an understudied area of research in these organisms). The experiments are, for the most part, well-described and well-performed. I have numerous comments that the authors should take into account in future versions of the manuscript, including suggestions for additional experiments that would improve confidence in key findings of the study.

Major comments

1. Figure 1a. The XFM approach is a potentially powerful way to detect iron in parasites. However the data here depict only a single image. Is this representative of multiple images? If so how many (e.g. how often do Zn and Ca overlap? How often do Zn and Fe not overlap?)? Some degree of quantification would improve the confidence in these data. The same comment applies for Figure 1h.

2. Figure 2. The data here depict hypersensitivity of Δ VIT parasites to exogenous iron. The data are striking, and are at the core of the key findings from the manuscript. Although they present evidence that VIT knockout was successful in Figure 1 (but note my query on Figure 1d below), the authors cannot rule out that a secondary genetic effect is causing the phenotype. The data would be strengthened by inclusion of a complementation (i.e. add-back) control to restore VIT expression in the Δ VIT strain, testing whether some of the key effects (e.g. hypersensitivity to exogenous iron, Figure 2c) are mitigated in a complemented strain.

3. Figure 4b-c. In Figure 4b, FAC and DFO appear to result in decreased VIT-HA abundance, but also of TOM40 abundance. A potential explanation then is that FAC and DFO treatment leads to an overall decrease in protein abundance in parasites. Can the authors clarify whether they normalized the quantification of their blots in Figure 4c relative to overall protein amount (or to TOM40 abundance)? If not, they should do so to account for this possibility.

4. Figure 5h and lines 456-457. “These results suggest that in the Δ VIT line, parasites upregulate Mtf at the protein level”. This is an interesting observation, but based on quantification of immunofluorescence images. As a more accurate means of quantification, the authors should conduct western blotting to measure the abundance of Mtf-HA in parental vs Δ VIT parasites.

Minor comments

5. Line 17. “iron forms dangerous oxygen radicals”. Strictly, iron leads to the formation of oxygen radicals

6. Line 22. “that iron is restricted to a compartment in the parasite that does not overlap with zinc”. Iron is likely present in other compartments as well, just not at such high concentrations. Consider re-wording to “the bulk of iron” or similar.

7. Line 38. “and virulent in multiple tissue types” – “is virulent”, although virulence describes effects on an entire organism rather than particular tissues. “Infectious in multiple tissue types” is perhaps a better expression.

8. Lines 87-88, “VIT expression is regulated changes in exogenous iron in *T. gondii*” – meaning not clear. “is regulated upon changes to exogenous iron”?

9. Figure 1d. What is the arrow depicting? The expected size of the gene? There appears to be a band for this still present in the Δ VIT strain.

10. Figure 2a and Lines 162-163. “forming fewer and much smaller plaques than the parental line”. The plaques are clearly smaller in excess FAC, but the data presented do not clearly show fewer plaques (and in fact, it appears there are more plaques in the Δ VIT strain in the 0 μ M FAC condition). Consider quantifying this, or rewording to state just “much smaller plaques”. As an additional note: “VIT” in the figure should presumably read Δ VIT?

11. Figure 2b and Line 178. “lack of VIT sensitises parasites to exogenous iron”. The key comparison here is the day 2 condition comparing Δ VIT to the Δ VIT+200 μ M FAC condition. Are the differences between these two conditions statistically significant?

12. Figure 2c. The differences here are striking, but I’m a bit unclear about exactly what is being measured. The text says “quantifying parasite fluorescence” – is this fluorescence of individual parasites? Or overall well fluorescence (i.e. of the population of parasites in the well, with well fluorescence indicative of the extent of parasite proliferation across the four days)? Based on the description in the methods, I think the latter, but this should be described more clearly in the results text. As a more general comment on this and other similar experiments in the manuscript – the y axis title states “parasite survival” (implying that what is being measured is whether the parasites are alive or dead), whereas what I think is being measured is parasite proliferation. The authors could be clearer about this.

13. Lines 286-288. “Dynamic localisation through the lytic cycle has previously been observed in proteins localizing to the vacuolar associated compartment (VAC) (Thornton et al., 2019; Warring et al., 2014)”. The cited studies examine the dynamic localisation of TgCRT. Did the authors ever check whether VIT colocalises with TgCRT?

14. Figure 3i. On my version of the manuscript, this figure has poor resolution and the gold particles were difficult to see.

15. Figure 5a. Given that the authors are considering whether VIT has a role in mediating the response to ROS in the presence of excess iron, the statistical analysis here could also compare the “+FAC” condition in nNeon vs Δ VIT parasites. In fact, the text in lines 407-409 seems to suggest that this was the statistical comparison being made – the indicated p value in the figure seems to be comparing the -FAC vs +FAC condition in Δ VIT parasites only. The authors should clarify this.

16. The data in Figure 5d suggest an increase abundance in catalase in the Δ VIT strain (and in WT parasites upon FAC treatment). However, the authors see an apparent decrease in catalase activity in FAC-treated Δ VIT parasites. This seems an odd result if the hypothesis is that parasites increase catalase expression/activity upon Fe-induced ROS. Do the authors have any idea what might be going on here? These data would be enhanced by additional approaches to measure the abundance of catalase protein in these parasites (e.g. by western blotting).

17. Figure 5e-f. These experiments use MitoSOX as a means of measuring ROS in mitochondria. These data assume that MitoSOX is measuring mitochondrial ROS in these parasites. Has any previous study used MitoSOX with *T. gondii* parasites? If not, the authors should include data that test whether the MitoSOX dye localises to the mitochondrion of the parasite.

18. Line 480 (“VIT is required for virulence” and Figure 6 legend title (“VIT is required ... for pathogenesis”). These are overstating the results - Δ VIT parasites have reduced virulence, so still have some virulence/pathogenesis. Consider rewording (“VIT contributes to...” or “is important for ...”)

19. Line 617-618. “suggests a level of metabolic flexibility in *T. gondii*.” I don’t understand the meaning here. Do the authors mean flexibility in iron storage (which is different to metabolism)?

20. Line 632. “we provide the first study of an iron transporter in *T. gondii*”. Although the data presented in the manuscript are consistent with VIT being an iron transporter, the study does not directly test the ability of VIT to transport iron. I think the authors should be a bit more cautious here: ‘...first study of a putative iron transporter ...’ perhaps?

21. Line 676. “Parasites were selected with 50 μ g/mL phleomycin”. The reason for this wasn’t apparent. Does the repair template encode a phleomycin resistance marker?

22. Line 714. Merck?

23. Lines 715 and 716. For the anti-CDPK and anti-CPL antibodies, consider citing the studies in which these were generated (if applicable).

Reviewer #3 (Remarks to the Author):

This is a manuscript from Aghabi et al. describing the functional characterization of the vacuolar iron transporter (VIT) in the apicomplexan parasite *Toxoplasma gondii*. The authors identify the localization of metal storage in this parasite. They then produce a Knock-out of the TgVIT gene in a type I strain. The characterization of the phenotypes leads to the identification of the mutant hypersensitivity to iron suggesting this protein has a role in iron detoxification. Surprisingly, these mutant parasites have also a growth disadvantage in iron-depleted environments. Gene expression is slightly changed in absence of TgVIT, particularly some pathways linked to iron are modulated in the mutant. The authors show that TgVIP expression and localization are dependent on the iron concentration although it is difficult for the reader to assess whether these variations have physiological relevance. The authors also show that the absence of TgVIP is linked to decreased virulence in a mouse model. This is an interesting subject since iron metabolism has not been investigated in this organism. However, the data presented here is only incremental compared to what was already described in other apicomplexan parasites such as *Plasmodium*. There is also a number of concerns that are listed below.

Major concerns

- The biochemical characterization of the transporter is missing. It has been performed for *Plasmodium* species using recombinant proteins or complementation of the yeast transporter CCC1. This is important to fully understand the affinity of the transporter for different metals and would allow restricting its role to iron detoxification as suggested by the authors.
- It is a standard practice in *T. gondii* to perform complementation of the mutant using an exogenic copy of the mutated gene. The main conclusions of this manuscript should be verified using a complemented strain to ensure that the phenotypes observed are solely due to the mutated gene.
- In some of the experiments, the authors did not quantify their observations. This is particularly the case for the plaque assays that should be assessed through quantification of the size and number of plaques (Figures 1e, 2a and S2c/d). This is also true for the overlap of fluorescence presented in Figures 1a and 1f.
- The localization of the VIT protein is confusing depending on the tag used. Although the authors show differential localization (unquantified) with the VAC compartment depending on the tag employed, they mainly used the HA-tagged strain for further studies. The electron microscopy pictures are scarce and hard to interpret without quantification. Did the author perform these experiments with both tags? How does that localization is influenced in presence of FAC or DFO?
- Evaluation of virulence of the strains in mice has been only produced once for 10 mice. The author should increase the number of repeats before concluding the virulence of these strains. A complemented strain is also warranted to ensure that the phenotype observed is due to the absence of the targeted gene.

Minor concern

- The increased size of the VAC compartment may not be directly linked to iron concentration but acidification or increased redox activity. The authors should tone down their conclusions about these experiments.

We thank the reviewers for their time and attention. In reference to their comments we have added several new experiments and techniques to the manuscript, including yeast complementation, cellular thermal shift assays, further quantification of plaque assays and a number of new IFAs. The additional work has more closely defined the growth phenotype of the Δ VIT parasites under normal and high iron growth conditions (**Fig. 2a, b and Fig. S3a**). We have also now examined the ability of TgVIT to complement a yeast mutant (**Fig. S2**), although this did not prove possible. To demonstrate direct iron binding of a small, transmembrane protein would be highly challenging so we optimised a cellular thermal shift assay (**Fig. 2j**) demonstrating a change in protein stability upon iron treatment, providing further evidence that VIT functions as an iron transporter in these cells. To address questions about the localisation of VIT, we now present several new panels in **Fig. 3**, including additional immunoEM images, overlap with a new marker (VP1) and the additional marker CRT (**Fig. S6c**). We also show that MitoSOX localises to the mitochondrion in *T. gondii* (**Fig. S8d**), allaying worries about its specificity and show that MIT is upregulated by western blotting (**Fig. 5i and j**).

We have also made several changes to the wording and language of the paper (specified below) and now feel that the paper is clearer and easier to follow.

Reviewer #1 (Remarks to the Author):

This manuscript investigates the role of a conserved organellar/vacuolar iron transporter VIT in storage/detoxification of iron in *Toxoplasma*. The rationale for this work is based on the fact that *Toxoplasma*, in contrast to its host cells, lacks the main iron storage protein ferritin. VIT was previously identified in plants and closely related *Plasmodium* and shown to be involved in transport/detoxification of iron. The structure of *Eucalyptus grandis* VIT has been solved. TgVIT was identified based on homology to its *Plasmodium* counterpart and shows significant conservation in residues important for iron binding. In this work, the authors generate various VIT null cells and show that these cells become hypersensitive to increased iron levels, although the nulls are viable. They show that increased sensitivity to exogenous iron in VIT nulls was due to production of excess ROS and this effect was reversed by addition of N-acetyl cysteine. Transient overexpression and epitope-tagging of VIT showed changes in its localisation from a single focus which became more dispersed through the lytic cycle. Colocalization of VIT with a vacuolar associated compartment marker suggests that this compartment substitutes for Fe storage systems that are used in other systems. Finally, VIT null cells have impaired survival in in vivo infectivity.

Iron-associated toxicity is an extremely interesting topic, not only for those interested in *Toxoplasma* biology, but also for higher eukaryotes – where many iron-associated disorders are lethal. While the experiments throughout the manuscript were done carefully with good quality data and rigorous statistical analyses, there are a few issues that I wish to highlight.

1. The evidence as presented for TgVIT conservation is weak. Fig 1b - The key amino acids shown does not reflect the full architecture and domain organisation of the VIT1 family across different species. Asp43 and Met80 involved in the coordination of metals is shown

but other key residues involved in iron transport are not shown. From an evolutionary perspective it is important to show how many TM segments are present in TgVIT1, for example are Gly 44, 69, 76 in EgVIT1 (important for iron transport activity) conserved? In closely related PfVIT1, there are conserved glutamate residues in the cytoplasmic metal binding domain. Are these conserved in TgVIT1? I will suggest replacing the alignment with a schematic highlighting all key conserved residues: # of TMDs, conserved Gly, Glu and Met.

We thank the reviewer for the suggestion, and have added a much more extensive alignment at **Fig. S1**, including the conserved TMDs as suggested. We find that beyond the conservation of D43 and M80 (all residue numbers here refer to EgVIT, where the crystal structure has been solved), we see conservation of all 6 of the Glu residues and Met149 which are involved in ion binding and required for complementation of Δ ccc1 in yeast (**Fig S1**, red, (Kato et al 2019)). Interestingly, although G44 and 76 are conserved, we don't see conservation of G69 which appears to be replaced by a D in the coccidia and an N in *Plasmodium* (**Fig. 1b** and **Fig. S1**, pale orange). G69 is expected to be involved in proton transfer, however appears to have a weaker role than the other conserved residues. It would be expected that the replacement of G with N or D would alter the transport ability, however in this paper we have not examined this in more depth.

2. All the data presented rely on the fact that VIT nulls do not express VIT mRNA and thus protein. Based on the schematic in Fig 1c, the Knockout construct should result in replacement of all exons with mNeonGreen. In Fig 1d, PCR validation is presented with primers flanking regions within the 5' and 3'- ends of the repair template. If KO is achieved, Δ VIT1 cells should show no TgVIT1 transcripts, but this is not the case in Fig 4f, see RNA-Seq suppl Table S1. Relative to parental cells, VIT1 transcripts (geneID TGME49_266800) are only depleted by 60% ($\log_2FC = -0.7653217$, $padj = 0.00026238$) suggesting that there is clear VIT transcription. This issue can be clarified by performing quantitative real time PCR of VIT transcripts in Parental vs Δ VIT cells and linking PCR using primers within the repair template and up/downstream flanks of the VIT1 locus – outside the repair template. The same holds true for all VIT null cells: mNeon (Fig 1), Δ ku80 mNeon, tdTomato, and DHFR. It is important to clarify which KO cell line was used for RNA-Seq and ensure that there are no VIT transcripts in all KO cells for all other analyses to be valid.

The residual transcripts seen in our Δ VIT line are due to the way we constructed the mutant, as we only replaced the coding region of the gene, we do see some transcripts from the UTRs, as shown below. As you can see, almost all reads from the Δ VIT line map to the 5' UTR region of the gene, with no mapping in the coding sequence. We do not expect that this would lead to the production of any functional protein. We have added the below diagram to **Fig. S7a** and added an explanatory note to **Table S1** to ensure that this is clear. Further, the primers used to validate the replacement of *vit* in both KO lines included one binding within the repair construct and the other in the genomic UTR sequence, ensuring that the repair construct was integrated where we expected.

3. The conclusion that VIT localises to the vacuolar compartment may be correct, but it is compromised by three major issues. (i) transient overexpression (OE): I did not find much utility of these data as they don't reflect endogenous levels of VIT and VIT did not show increased expression in either low or excess iron.

Secondly, the authors admit that VIT::Ty OE is toxic as these cells did not grow for prolonged periods. The conclusions from these data could be due to unavoidable excess protein that swamp the cells. The "detail" highlighted in Fig S4 is not convincing. I see no significant colocalization by eye. Most of the membrane localised outline seen in Ty is not present in HA. Myc and HA are convincing.

We agree that the overexpression of VIT does not contribute to the manuscript and have removed these figures.

(ii) in the EM in Fig 3i, some VIT in (i) is in the lumen of that vacuole. Isn't VIT a transmembrane protein?

This is correct, VIT has 4 predicted transmembrane domains. The reason that the gold appears to be in the lumen is likely a combination of the size of the primary and secondary antibodies used (meaning that the signal may appear up to 30 nm from the antigen (Hermann et al, 1996, *Histochemistry and Cell Biology*)), and the thin sections of a 3D structure. We have included more relevant EM images from an independent experiment (**Fig. 3g**) in the manuscript, including those where the signal is clearly very close to the membrane.

(iii) the authors argue that PfVIT may be mis-localised due to addition of GFP-tags. This explanation holds true for their data. One way to resolve this will be to knock-in HA- or Myc- epitope-tagged versions and show that these can rescue iron toxicity in VIT nulls.

We have attempted complementation using a number of strategies however were unsuccessful in generating a stable line. We do feel that the localisation we see is likely to be correct based on the fact that two independent tags (HA and Myc) show the same localisation, and even overexpression with a Ty tag shows localisation to the PLVAC. Further, small epitope tags are less likely to cause misslocalisation than the bulky GFP tags (e.g. Wichers et al 2019, mBio). In other organisms such as yeast and plants, VIT is localised to acidic vacuoles which fits with the localisation that we see, as the PLVAC is known to be acidified (Stasic et al 2019). Further, from our results here we see that iron (in

extracellular parasites) is localised mostly to a distinct punctate, which does not appear to reflect the ER.

While we cannot rule out the idea that our tags are causing misslocalisation, we have attempted to moderate our language in the discussion to ensure that these limitations are clear.

4. While the title is not an overstatement, the evidence present is indirect. Direct and unequivocal evidence for intracellular iron detoxification could be provided by (i) showing that TgVIT binds iron, and (ii) fractionation of organellar vs cytosolic and measuring iron content in these compartments relative to total levels in null vs parental cells.

We agree that the evidence presented here that VIT is an iron transporter is mostly indirect. Unfortunately we are unable to generate sufficient material to measure iron in cytosolic and vacuolar fractions (to measure iron in whole cells requires around 1×10^9 parasites and there is no method for vacuolar fractionation in *T. gondii*). To account for this we provide new data using cellular thermal shift assays (CETSA), presented in **Fig. 2j**. Here we show that incubation of the parasites with high levels of iron changes the thermal solubility of VIT (but not our control protein, CDPK1 which binds the divalent cation Ca^{2+}). This experiment does not directly show that VIT binds to iron, such an assay is beyond the scope of this paper, however it provides strong supporting evidence, along with the other results that we present here, that VIT is involved directly in iron detoxification. Further the conservation of key residues and the function of VIT in *Plasmodium*, plants and yeast all support our conclusions for the role of VIT in *T. gondii*.

Minor points:

Fig 1a: Needs a scale bar to appreciate the localisation/dispersal of the different metals visualised across the panels shown. How many parasites were looked at per metal? Representative images of the number cells visualised should be stated.

We have added a scale bar as requested. Unfortunately, given the limitations of XFM we were only able to visualise 3-4 parasites/condition. We have added this information to results section and added the Pearson's correlation coefficient to the images shown.

Fig 1d. arrow indicates?

The arrow indicates the band of expected size, other bands are likely unspecific. This has been described in the figure legend.

Fig 2a: Top panel should be Δ VIT not VIT, for consistency. I would have liked to see Parental and Δ VIT experiments at 50 and 200 micro molar not 10 and 500. Note: Δ VIT::DHFR cells are done at 50 and 200 micro molar. It makes comparisons consistent.

This is an excellent point and we have performed new experiments (**Fig. 2a**) at the suggested concentrations and quantified the results of these plaque assays (**Fig. 2b and S3a**).

Fig 2b: 200 micro molar FAC does not affect survival of mNeon (2b) but what does the plague assay look like?

We have included the image (**Fig. 2a**) and quantification (**Fig. 2b and S3a**) of the number and area of plaques present in the mNeon line upon 200 μ M treatment.

To what extent are the iron levels used in the in vitro experiments representative of physiological conditions. I found the choice of FAC concentrations at times excessive, 5 mM in RNA-Seq experiment and Western blot Fig 4 contrasting 200 μ M in Fig 2.

The iron levels used in this paper are above normal physiological levels. This was a deliberate choice, as we wanted to make sure we were overwhelming the host cell buffering responses to ensure that the intracellular parasites faced differing levels of iron. However, Δ VIT parasites have a significant phenotype in infection (**Fig. 6a**) which demonstrates that these effects are likely important *in vivo*.

Fig S2b – It appears the gel image has been contrast-enhanced to mask a band in the parental. Please present original image

We have replaced this panel (**Fig. S4b**) with a new PCR and gel (see below) which confirmed the integration of the cassette (expected PCR product: 691bp). We saw no bands in our negative control.

Reviewer #2 (Remarks to the Author):

In this manuscript, Aghabi and colleagues describe the role for a putative vacuolar iron transporter (VIT) in iron storage and homeostasis in *Toxoplasma gondii* parasites. The authors demonstrate that VIT has a critical role in mediating the parasite's response to iron detoxification, with loss of the transporter likely leading to increased production of reactive oxygen species in the parasite. They demonstrate a role for the transporter in mediating both parasite proliferation and virulence. They also present some evidence that the abundance of the VIT protein is regulated by exogenous iron levels, although the data on this are not entirely convincing (see comments below). Overall the manuscript breaks important new ground in understanding the responses of an intracellular parasite to the iron levels it encounters in its environment, and develops some powerful new approaches for analysing iron homeostasis in the parasite (an understudied area of research in these organisms). The experiments are, for the most part, well-described and well-performed. I have numerous comments that the authors should take into account in future versions of the manuscript,

including suggestions for additional experiments that would improve confidence in key findings of the study.

Major comments

1. Figure 1a. The XFM approach is a potentially powerful way to detect iron in parasites. However the data here depict only a single image. Is this representative of multiple images? If so how many (e.g. how often do Zn and Ca overlap? How often do Zn and Fe not overlap?)? Some degree of quantification would improve the confidence in these data. The same comment applies for Figure 1h.

We have added the Pearsons correlation to the XFM images. As said above, the images are representative of 3-4 parasites/strain. Given the limitations of beamtime available, it was not possible to capture enough images for accurate quantification. We have made these limitations clear in the text to ensure that we do not draw unwarranted conclusions.

2. Figure 2. The data here depict hypersensitivity of Δ VIT parasites to exogenous iron. The data are striking, and are at the core of the key findings from the manuscript. Although they present evidence that VIT knockout was successful in Figure 1 (but note my query on Figure 1d below), the authors cannot rule out that a secondary genetic effect is causing the phenotype. The data would be strengthened by inclusion of a complementation (i.e. add-back) control to restore VIT expression in the Δ VIT strain, testing whether some of the key effects (e.g. hypersensitivity to exogenous iron, Figure 2c) are mitigated in a complemented strain.

Despite multiple attempts, we were unable to make a stable complemented line. However, in the paper we use two independent KOs (Δ VIT:mNeon and Δ VIT:DHFR) which we show to have the same phenotype (e.g. **Fig. 2a, b, S3a and Fig S4c, d**), providing strong evidence that the effects we see are due to the absence of VIT. Further, from our RNAseq data, we also do not see any evidence of other genetic manipulations in our line. Also the new data we provide in **Fig. 2j** which gives evidence of iron binding to VIT-HA also supports our proposed role of VIT in *T. gondii*.

3. Figure 4b-c. In Figure 4b, FAC and DFO appear to result in decreased VIT-HA abundance, but also of TOM40 abundance. A potential explanation then is that FAC and DFO treatment leads to an overall decrease in protein abundance in parasites. Can the authors clarify whether they normalized the quantification of their blots in Figure 4c relative to overall protein amount (or to TOM40 abundance)? If not, they should do so to account for this possibility.

In these blots we did normalise to TOM40 levels and have added this in the text as below:

“Interestingly, removal of iron by DFO also led to a decrease ($p = 0.032$, one sample t test) in VIT-HA levels (normalised to the mitochondrial protein TOM40) despite no change in RNA levels.”

As the reviewer notes, upon both FAC and DFO treatment we do see generally lower levels of protein, we mention this in the discussion, however the mechanisms controlling this change in expression are beyond the scope of this paper.

4. Figure 5h and lines 456-457. “These results suggest that in the Δ VIT line, parasites upregulate Mtf at the protein level”. This is an interesting observation, but based on quantification of immunofluorescence images. As a more accurate means of quantification, the authors should conduct western blotting to measure the abundance of Mtf-HA in parental vs Δ VIT parasites.

*Please note we have renamed Mtf to MIT (Mitochondrial iron transporter) to avoid confusion with mitofusins

Thank you for the suggestion, we were able to visualise MIT-HA by western blotting using film and have included the new data in **Fig. 5i** and **j**. We see a reproducible increase in MIT-HA compared to the parental tagged line.

Minor comments

5. Line 17. “iron forms dangerous oxygen radicals”. Strictly, iron leads to the formation of oxygen radicals

This has been corrected as below:

“iron leads to the formation of dangerous oxygen radicals:

6. Line 22. “that iron is restricted to a compartment in the parasite that does not overlap with zinc”. Iron is likely present in other compartments as well, just not at such high concentrations. Consider re-wording to “the bulk of iron” or similar.

This has been corrected as below:

“We show that the bulk of iron is restricted to a compartment in the parasite that does not overlap with zinc”

7. Line 38. “and virulent in multiple tissue types” – “is virulent”, although virulence describes effects on an entire organism rather than particular tissues. “Infectious in multiple tissue types” is perhaps a better expression.

This has been corrected as below:

“...able to infect most warm-blooded species, almost all nucleated cells and able to replicate in multiple tissue types.”

8. Lines 87-88, “VIT expression is regulated changes in exogenous iron in T. gondii” – meaning not clear. “is regulated upon changes to exogenous iron”?

This has been corrected as below:

“Further, VIT expression is regulated in response to changes in exogenous iron levels.”

9. Figure 1d. What is the arrow depicting? The expected size of the gene? There appears to be a band for this still present in the Δ VIT strain.

Yes, the arrow represents the size of the PCR product to be expected when the gene is present (as shown on **Fig. 1c**), the asterisks show unspecific bands. We have replaced the gel image to make it clear that there is no band at the appropriate size in the Δ VIT strain. Please note, in the published version we have cropped the image to remove the primer dimers.

10. Figure 2a and Lines 162-163. “forming fewer and much smaller plaques than the parental line”. The plaques are clearly smaller in excess FAC, but the data presented do not clearly show fewer plaques (and in fact, it appears there are more plaques in the Δ VIT strain in the 0 μ M FAC condition). Consider quantifying this, or rewording to state just “much smaller plaques”. As an additional note: “VIT” in the figure should presumably read Δ VIT?

This panel has been replaced with new plaque assays performed at 0, 50 and 200 μ M FAC, including quantification of the plaque area (**Fig. 2a and b, Fig. S3a**), at the request of reviewer 1. We have reworded the text to that below:

*We found that Δ VIT parasites were noticeably more sensitive to excess FAC, forming fewer and significantly smaller plaques ($p < 0.001$ at 200 μ M FAC, t test) than the parental line (**Fig. 2a and b, Fig. S3a**).*

11. Figure 2b and Line 178. “lack of VIT sensitises parasites to exogenous iron”. The key comparison here is the day 2 condition comparing Δ VIT to the Δ VIT+200 μ M FAC condition. Are the differences between these two conditions statistically significant?

Yes, this comparison is significantly different ($p = 0.0017$, t -test corrected with Holm-Sidak) and this has been added to the text as below:

Addition of FAC exacerbated this phenotype, Δ VIT parasites were significantly ($p = 0.001$, t test, Holm-Sidak corrected) outcompeted by two days post infection in the presence of excess iron and were almost undetectable by four days post infection.

12. Figure 2c. The differences here are striking, but I’m a bit unclear about exactly what is being measured. The text says “quantifying parasite fluorescence” – is this fluorescence of individual parasites? Or overall well fluorescence (i.e. of the population of parasites in the well, with well fluorescence indicative of the extent of parasite proliferation across the four days)? Based on the description in the methods, I think the latter, but this should be described more clearly in the results text. As a more general comment on this and other similar experiments in the manuscript – the y axis title states “parasite survival” (implying that what is being measured is whether the parasites are alive or dead), whereas what I think is being measured is parasite proliferation. The authors could be clearer about this.

The reviewer is correct here and we apologise for this oversight, we have explained our assay more thoroughly in the results (as below) and have changed the label on all of the relevant figures to '% parasite proliferation'.

To quantify the iron hypersensitivity of the Δ VIT parasites, we infected host cells in 96-well plates with mNeon or Δ VIT parasites and treated with increasing concentrations of FAC for four days before measuring the fluorescence of each well using a plate reader, normalized to untreated wells (Fig. 2d). This allowed us to quantify the degree of parasite proliferation in the presence of a range of iron concentrations.

13. Lines 286-288. "Dynamic localisation through the lytic cycle has previously been observed in proteins localizing to the vacuolar associated compartment (VAC) (Thornton et al., 2019; Warring et al., 2014)". The cited studies examine the dynamic localisation of TgCRT. Did the authors ever check whether VIT colocalises with TgCRT?

We have performed localisation with another marker of the PLVAC, VP1 (Miranda et al, 2010), and show overlap in extracellular parasites (Fig. 3d). We also examined CRT, through transient overexpression. Interestingly, we did not see overlap between VIT-HA and CRT-GFP/mCherry (Fig. S6c), however this may be due to the overexpression of CRT. Further, the degree of overlap between proteins believed to localise to the PLVAC is complex, e.g. in intracellular parasites CRT and CPL overlap but VP1 and CRT only partially overlap (Warring et al 2014). We believe that the composition of the PLVAC is complex, consisting of a number of highly dynamic markers with differing degrees of overlap. We have attempted to explain this more clearly in the text, however a full understanding of the composition of the PLVAC will require further work outside the scope of this paper.

14. Figure 3i. On my version of the manuscript, this figure has poor resolution and the gold particles were difficult to see.

We have replaced the images (and added further images) to **Figure 3g**. To ensure that the gold particles are clear, we have made the images larger and added arrowheads.

15. Figure 5a. Given that the authors are considering whether VIT has a role in mediating the response to ROS in the presence of excess iron, the statistical analysis here could also compare the "+FAC" condition in nNeon vs Δ VIT parasites. In fact, the text in lines 407-409 seems to suggest that this was the statistical comparison being made – the indicated p value in the figure seems to be comparing the -FAC vs +FAC condition in Δ VIT parasites only. The authors should clarify this.

Our apologies for the lack of clarity, we have amended the figure to add this important comparison and altered the text in the results section as shown below:

However, upon FAC treatment ROS levels were significantly ($p = 0.03$, one way ANOVA with Sidak's correction) higher in the Δ VIT strain than in the parental and significantly raised ($p = 0.004$, one way ANOVA with Sidak's correction) compared to the untreated Δ VIT line (Fig. 5a).

16. The data in Figure 5d suggest an increase abundance in catalase in the Δ VIT strain (and in WT parasites upon FAC treatment). However, the authors see an apparent decrease in catalase activity in FAC-treated Δ VIT parasites. This seems an odd result if the hypothesis is that parasites increase catalase expression/activity upon Fe-induced ROS. Do the authors have any idea what might be going on here? These data would be enhanced by additional approaches to measure the abundance of catalase protein in these parasites (e.g. by western blotting).

We agree that the apparent drop in catalase activity (we did not examine protein abundance) in FAC-treated Δ VIT parasites was surprising, although this drop was not significant. However, high levels of oxidative stress have been shown to inhibit catalase activity in yeast (Martins and English, 2014) and plants (Shim et al, 2003). It is possible that the high levels of ROS seen in FAC-treated Δ VIT parasites are overwhelming the defences and leading to the death of a proportion of the cells, leading to an apparent drop in enzyme activity. Unfortunately, after consulting a number of sources we were not able to obtain a *Toxoplasma* anti-catalase antibody. It appears that stocks have been depleted since it was published and no more has been made. However, we feel that the interesting data point here is the upregulation of catalase activity under normal growth conditions. We have modified the text as below to make this clear.

Interestingly, we saw an apparent decrease in catalase activity (although this was not significant) in the Δ VIT line upon FAC treatment. It has previously been reported that oxidative stress can decrease catalase activity in plants (Shim et al., 2003) and yeast (Martins and English, 2014) which could explain why we only see this inhibition of activity in the Δ VIT strain, which is under the highest levels of oxidative stress (Fig. 5a).

17. Figure 5e-f. These experiments use MitoSOX as a means of measuring ROS in mitochondria. These data assume that MitoSOX is measuring mitochondrial ROS in these parasites. Has any previous study used MitoSOX with *T. gondii* parasites? If not, the authors should include data that test whether the MitoSOX dye localises to the mitochondrion of the parasite

We agree, we have included images showing that MitoSOX colocalises with the well established mitochondrial dye MitoTracker in live cells in **Fig. S8d** and added this information to the text as below:

To examine if excess iron alters mitochondrial ROS (mROS) accumulation, we stained parasites with the mitochondrial-specific ROS probe MitoSOX (Fig. 5e and f) after confirming that MitoSOX localised to the parasite's mitochondrion (as defined by mitotracker) in live cells (Fig. S8d).

18. Line 480 ("VIT is required for virulence" and Figure 6 legend title ("VIT is required ... for pathogenesis"). These are overstating the results - Δ VIT parasites have reduced virulence, so still have some virulence/pathogenesis. Consider rewording ("VIT contributes to..." or "is important for ...")

We have changed this heading as suggested to "VIT contributes to virulence *in vivo*" and ensured that we do not overstate the results in the text.

19. Line 617-618. “suggests a level of metabolic flexibility in *T. gondii*.” I don’t understand the meaning here. Do the authors mean flexibility in iron storage (which is different to metabolism)?

We have changed this to:

*“and suggests a level of flexibility in iron compartmentalization *T. gondii*.”*

20. Line 632. “we provide the first study of an iron transporter in *T. gondii*”. Although the data presented in the manuscript are consistent with VIT being an iron transporter, the study does not directly test the ability of VIT to transport iron. I think the authors should be a bit more cautious here: ‘...first study of a putative iron transporter ...’ perhaps?

While we agree that in this work we have not shown iron transport activity, given the totality of evidence presented in this work, as well as the work from the closely related *Plasmodium* VIT transporter, we are confident that this protein is acting as an iron transporter in this context. However, we have modulated our language in this paragraph as below:

*In conclusion, we provide the first study of a proposed iron transporter in *T. gondii*. We show that it has a role, although is not essential, for growth under normal conditions, but is vital in detoxification of excess iron.*

21. Line 676. “Parasites were selected with 50 µg/mL phleomycin”. The reason for this wasn’t apparent. Does the repair template encode a phleomycin resistance marker?

We thank the reviewer for this comment. The plasmid that contains the Cas9 and sgRNA contains the PhleoR resistance marker. We have clarified this in the materials and methods and cited our first use of this plasmid.

22. Line 714. Merck?

Apologies, this has been corrected

23. Lines 715 and 716. For the anti-CDPK and anti-CPL antibodies, consider citing the studies in which these were generated (if applicable).

These references have been added.

Reviewer #3 (Remarks to the Author):

This is a manuscript from Aghabi et al. describing the functional characterization of the vacuolar iron transporter (VIT) in the apicomplexan parasite *Toxoplasma gondii*. The authors

identify the localization of metal storage in this parasite. They then produce a Knock-out of the TgVIT gene in a type I strain. The characterization of the phenotypes leads to the identification of the mutant hypersensitivity to iron suggesting this protein has a role in iron detoxification. Surprisingly, these mutant parasites have also a growth disadvantage in iron-depleted environments. Gene expression is slightly changed in absence of TgVIT, particularly some pathways linked to iron are modulated in the mutant. The authors show that TgVIP expression and localization are dependent on the iron concentration although it is difficult for the reader to assess whether these variations have physiological relevance. The authors also show that the absence of TgVIP is linked to decreased virulence in a mouse model. This is an interesting subject since iron metabolism has not been investigated in this organism. However, the data presented here is only incremental compared to what was already described in other apicomplexan parasites such as Plasmodium. There is also a number of concerns that are listed below.

Major concerns

- The biochemical characterization of the transporter is missing. It has been performed for Plasmodium species using recombinant proteins or complementation of the yeast transporter CCC1. This is important to fully understand the affinity of the transporter for different metals and would allow restricting its role to iron detoxification as suggested by the authors.

We include the complementation of the yeast now in **Figure S2**. We did not see much ability of TgVIT to complement the iron sensitivity of the $\Delta ccc1$ yeast strain under these conditions. Interestingly, we do find that while expression of a shorter form of VIT is toxic in wildtype yeast, this toxicity is abrogated in the $\Delta ccc1$ strain, suggesting a some function for TgVIT in yeast. It is possible that the expression level or localisation of TgVIT was not optimal in these experiments, however we feel that further investigation of this is beyond the scope of the paper. In addition, given the totality of our other results (e.g. the change of iron localisation and quantity, the hypersensitivity and increased ROS of the knockout) we feel confident in our major conclusions.

- It is a standard practice in *T. gondii* to perform complementation of the mutant using an exogenic copy of the mutated gene. The main conclusions of this manuscript should be verified using a complemented strain to ensure that the phenotypes observed are solely due to the mutated gene.

Despite numerous attempts and different strategies, we have been unable to complement our KO. We have found that overexpression of VIT is highly toxic to the parasite and so isolating a complemented clone has not been possible. However, to mitigate this valid concern we have made two independent KOs (replacing VIT with mNeonGreen or DHFR) and see identical phenotypes (e.g. **Fig. 2a** and **Fig. S2a**). Further, we have carefully examined our RNAseq data and find no evidence of any other genetic rearrangements in our strain.

- In some of the experiments, the authors did not quantify their observations. This is particularly the case for the plaque assays that should be assessed through quantification of the size and number of plaques (Figures 1e, 2a and S2c/d). This is also true for the overlap of fluorescence presented in Figures 1a and 1f.

Of course, we have now added new quantification (of numbers and area) for the plaque assays in **Fig. 2b** and **S1a**. We have also added the Pearson's correlation coefficient to **Fig. 1a** and **h**.

- The localization of the VIT protein is confusing depending on the tag used. Although the authors show differential localization (unquantified) with the VAC compartment depending on the tag employed, they mainly used the HA-tagged strain for further studies. The electron microscopy pictures are scarce and hard to interpret without quantification. Did the author perform these experiments with both tags? How does that localization is influenced in presence of FAC or DFO?

We thank the reviewer for their comments. We have addressed this in a largely new figure 3. We present additional immunoEM images (**Fig. 3e**). We also find that incubation of extracellular parasites in buffer A (as described in materials and methods) results in very similar localizations for both the HA and Myc tags. We also show overlap with the additional PLVAC marker VP1 (**Fig. 3f**). These additional data, in combination with the quantified changes in localisation upon FAC/DFO treatment (**Fig. 4e and f**) will hopefully make the localisation clearer in the manuscript.

- Evaluation of virulence of the strains in mice has been only produced once for 10 mice. The author should increase the number of repeats before concluding the virulence of these strains. A complemented strain is also warranted to ensure that the phenotype observed is due to the absence of the targeted gene.

The *in vivo* experiment was performed twice, using different numbers of parasites, and we saw the same trend in both experiments. Statistical significance analysis also clearly showed that the virulence in the mutant parasites was significantly reduced compared to WT strain. Therefore, we do not feel that it is necessary to repeat this assay further at this time.

Minor concern

- The increased size of the VAC compartment may not be directly linked to iron concentration but acidification or increased redox activity. The authors should tone down their conclusions about these experiments.

We have added this suggestion to the text, please see below:

Although we cannot rule out the role of other cellular changes, such as the redox state of the cell on VAC morphology, these results suggest the VAC is altered by excess iron through the actions of VIT,

REVIEWER COMMENTS

Reviewer #1 (Remarks to the Author):

The authors have responded appropriately to my comments with text changes and new data that I find satisfactory. Overall, solid evidence has been made for VIT as an iron transporter in Toxo. I wish to congratulate them on this interesting study.

I don't require a response but the question remains in my mind. It beats me why it is possible possible to knockout VIT using mNeonGreen and DHFR, but impossible to knock-in wild type VIT back into the same locus by CRISPR.

Reviewer #2 (Remarks to the Author):

Overall, the study by Aghabi and colleagues provides some valuable new insights into iron biology in the intracellular parasite *Toxoplasma gondii*. In their revision, the authors have addressed my major criticisms of the original manuscript, in part by including some new data. These mostly improve the manuscript (although, as noted below, I am unconvinced by the CETSA data). I have several further comments for the authors' consideration.

Major

Figure 2j. The western blots of the VIT-HA CETSA experiment are very messy, with a lot of background and dissimilar intensities between the untreated and FAC-treated conditions. The quantification also seems a bit odd, with soluble protein increasing in the temperatures up to 48°C before decreasing again. I'm not entirely convinced by these data.

Figure 5f. How were 'mitoSOX positive' cells identified? From the flow cytometry data in Fig 5e, it seems more like the main population gets a bit brighter, the 'mito-SOX high' population fragments in the FAC+ condition of the Δ VIT parasites, and that there are considerably more mito-SOX high parasites in the

mNeon +FAC condition than in the Δ VIT-FAC condition. None of these observations seem clearly captured by the graph in Figure 5f. The authors should define and justify their gating strategies here.

As a general comment on a few of the figure panels – the curve fitting in the graphs is often not a good approximation of the data points. This is most notable in Figures 2d, 2j (VIT-HA), 5b, and S5c, and affects things like calculations of the EC50 values in 2d and 5b, and the 50% protein solubility calculations in 2j.

Minor

Figure 1 legend. “Pearson”, “mislocalised”. “No change was seen in Zn, Ca, P or S” – no change compared to what? Intra vs extra-cellular or WT vs KO?

Figure S2. Perhaps define sVIT in the legend. As a more general question, I don’t understand the rationale behind testing the N-terminally truncated VIT.

p. 5. “with, or without, with 200 μ M FAC” – meaning not clear

Figure S4. Provide some indication of the number of times these experiments were repeated.

p. 16. “we saw an apparent accumulation of MIT-HA in the Δ VIT line” – not clear what is meant by accumulation. An increase in abundance? Accumulation to the mitochondrion?

p. 16. “This was confirmed by western blotting of MIT-HA (Fig. S9c and d)” – I think these data are Fig. 5i-j.

Figure 6 legend. “VIT is required for parasite survival in macrophages” – the authors have not shown that VIT is required (since they haven’t directly tested whether the parasites that have infected macrophages remain viable). Consider rewording to ‘is important for/contributes to’ parasite survival (and also in the discussion). For a similar reason, I don’t think ‘parasite survival’ is an appropriate y axis label in Fig 6c.

p. 21. “Trypanosoma”

Reviewer #3 (Remarks to the Author):

This is a revised manuscript from Aghabi et al. entitled "The vacuolar iron transporter mediates iron detoxification in *Toxoplasma gondii*". The authors did answer most of the previous concerns. It remains the critical complementation experiment.

Major concern

Complementation of the mutant using an exogenous copy of the TgVIT gene is critical to ensure that the phenotypes observed solely depend on the absence of the gene. The authors did not explain what the "numerous attempts and different strategies" that were attempted were. It seems that they tried to overexpress the protein ("We have found that overexpression of VIT is highly toxic to the parasite and so isolating a complemented clone has not been possible"). However, due to the toxicity of overexpressing the TgVIT protein, a complementation strategy using the TgVIT promoter to drive the exogenous copy seems warranted. The main conclusions of this manuscript should be verified using this complemented strain.

REVIEWER COMMENTS

Reviewer #1 (Remarks to the Author):

The authors have responded appropriately to my comments with text changes and new data that I find satisfactory. Overall, solid evidence has been made for VIT as an iron transporter in Toxo. I wish to congratulate them on this interesting study.

I don't require a response but the question remains in my mind. It beats me why it is possible to knockout VIT using mNeonGreen and DHFR, but impossible to knock-in wild type VIT back into the same locus by CRISPR.

We thank the reviewer for their comments, and agree wholeheartedly with their second point. We are pleased to say we finally succeeded in complementing the line! Please see more details in the reply to reviewer 3.

Reviewer #2 (Remarks to the Author):

Overall, the study by Aghabi and colleagues provides some valuable new insights into iron biology in the intracellular parasite *Toxoplasma gondii*. In their revision, the authors have addressed my major criticisms of the original manuscript, in part by including some new data. These mostly improve the manuscript (although, as noted below, I am unconvinced by the CETSA data). I have several further comments for the authors' consideration.

Major

Figure 2j. The western blots of the VIT-HA CETSA experiment are very messy, with a lot of background and dissimilar intensities between the untreated and FAC-treated conditions. The quantification also seems a bit odd, with soluble protein increasing in the temperatures up to 48°C before decreasing again. I'm not entirely convinced by these data.

We agree that these western blots are not ideal and have moved this data to the supplemental. As VIT is expressed at a low level, this experiment was very challenging to perform however we feel it is best to include it.

Figure 5f. How were 'mitoSOX positive' cells identified? From the flow cytometry data in Fig 5e, it seems more like the main population gets a bit brighter, the 'mito-

SOX high' population fragments in the FAC+ condition of the Δ VIT parasites, and that there are considerably more mito-SOX high parasites in the mNeon +FAC condition than in the Δ VIT-FAC condition. None of these observations seem clearly captured by the graph in Figure 5f. The authors should define and justify their gating strategies here.

To improve the representation of this data, we have added a new panel in **figure S8e** showing the geometric mean fluorescence intensity (MFI) of MitoSOX signal from the experiments. We consider that adding both graphs helps us capture some of the observations from our data which were missed from figure 5f and thank the reviewer for their comments.

As a general comment on a few of the figure panels – the curve fitting in the graphs is often not a good approximation of the data points. This is most notable in Figures 2d, 2j (VIT-HA), 5b, and S5c, and affects things like calculations of the EC50 values in 2d and 5b, and the 50% protein solubility calculations in 2j.

In Figure 5b the R^2 for Δ VIT is 0.81 and for Δ VIT+NAC is 0.7. We agree that the results in Figure S5c are not optimal, but we chose to keep the same conditions across all conditions to ensure that the comparisons that we made were fair. By changing the regression equation of figure S5c we did not materially affect the results.

Multiple approaches were attempted, however the curves presented fit the data the best (based on R^2 values). To ensure that we present our certainty of our results, we have documented the 95% C.I of all our calculations and provide the raw data.

Minor

Figure 1 legend. "Pearson", "mislocalised". "No change was seen in Zn, Ca, P or S" – no change compared to what? Intra vs extra-cellular or WT vs KO?

Thank you, we have corrected this to the below:

X-ray fluorescence microscopy examining elemental composition of intracellular Δ VIT parasites. No change was seen in Zn, Ca, P or S in Δ VIT cell compared to intracellular parental parasites, however Fe appeared potentially mislocalised.

Figure S2. Perhaps define sVIT in the legend. As a more general question, I don't understand the rationale behind testing the N-terminally truncated VIT.

This has been done, deletion of the N-terminal appears to improve the ability of PfVIT to complement the dCCC1 strain (Slavic et al, 2016) and we have amended the text in the results section to explain this:

Expression of full-length VIT in wild type yeast appeared toxic. We also tested a N-terminal truncation (sVIT₆₃₋₃₁₃) as this truncation of PfVIT was shown to successfully complement Δ ccc1 (Slavic et al., 2016). However, expression of either VIT construct did not complement the iron-hypersensitivity of Δ ccc1, although toxicity was abrogated. This contrasts with Plasmodium VIT (Slavic et al., 2016) and may be due to differences in expression or codon usage.

p. 5. “with, or without, with 200 μ M FAC” – meaning not clear

Thank you, we have corrected this oversight as below:

cultured with, or without, 200 μ M FAC

Figure S4. Provide some indication of the number of times these experiments were repeated.

These experiments were performed three times, we have added this to the figure legend:

Plaque assays showing increased sensitivity of the Δ VIT::DHFR_{TS} line to excess ferric ammonium citrate (FAC) (c) and ferrous ammonium sulphate (FAS) (d) at the indicated concentration compared to the parental parasite line. Representative plaque assays from three independent replicates.

p. 16. “we saw an apparent accumulation of MIT-HA in the Δ VIT line” – not clear what is meant by accumulation. An increase in abundance? Accumulation to the mitochondrion?

Our apologies, we have clarified this in the text as below:

Interestingly, we saw an increase in MIT-HA fluorescence in the mitochondrion in the Δ VIT line. We quantified this from immunofluorescence images and saw an increase in MIT-HA staining in the mitochondrion of the Δ VIT line compared to TOM40 (as an internal control) (Fig. 5h). An increase in total protein was confirmed by western blotting of MIT-HA (Fig. 5i and j). These results suggest that in the Δ VIT line, parasites upregulate MIT at the protein level, perhaps in an attempt to remove iron from the cytosol.

p. 16. “This was confirmed by western blotting of MIT-HA (Fig. S9c and d)” – I think these data are Fig. 5i-j.

Thank you, this has been corrected

Figure 6 legend. “VIT is required for parasite survival in macrophages” – the authors have not shown that VIT is required (since they haven’t directly tested whether the parasites that have infected macrophages remain viable). Consider rewording to ‘is important for/contributes to’ parasite survival (and also in the discussion). For a similar reason, I don’t think ‘parasite survival’ is an appropriate y axis label in Fig 6c.

We agree and we have changed the y axis label to “% parasite florescence” and have modified the figure legend to:

Figure 6. VIT contributes to parasite survival in macrophages and for pathogenesis *in vivo*

p. 21. “Trypanosoma”

Thank you, this has been corrected

Reviewer #3 (Remarks to the Author):

This is a revised manuscript from Aghabi et al. entitled “The vacuolar iron transporter mediates iron detoxification in *Toxoplasma gondii*”. The authors did answer most of the previous concerns. It remains the critical complementation experiment.

Major concern

Complementation of the mutant using an exogenous copy of the TgVIT gene is critical to ensure that the phenotypes observed solely depend on the absence of the gene. The authors did not explain what the “numerous attempts and different strategies” that were attempted were. It seems that they tried to overexpress the protein (“We have found that overexpression of VIT is highly toxic to the parasite and so isolating a complemented clone has not been possible”). However, due to the toxicity of overexpressing the TgVIT protein, a complementation strategy using the TgVIT promoter to drive the exogenous copy seems warranted. The main conclusions of this manuscript should be verified using this complemented strain.

Thank you, since receiving these reviews we have attempted 8 (further) times to complement the Δ VIT strain (using a variety of strategies) and were finally successful as shown below (and if figure S3).

We succeeded by putting *vit*, under the control of the native promoter, into the *uprt* locus as described in the materials and methods. This resulted in approximately double the expression compared to the wild type levels (**Fig. S3c**). We believe this is probably due to the *dhfr* 3' UTR, but may also be related to the wider genomic context. Complementation of the Δ VIT line led to a partial rescue of the iron hypersensitivity phenotype (**Fig. 2a, b, c and e**) by plaque assay and by fluorescent growth assay. We have discussed these results in the text, it is possible that full complementation would require the correct 3' UTR and genomic context but we feel that the results we generated support the conclusions of the paper.

We are very pleased that we were able to complete this part of the puzzle and we thank the reviewers for incentivising us to perform these important experiments.

REVIEWERS' COMMENTS

Reviewer #2 (Remarks to the Author):

The authors have addressed my concerns from the previous submission. I have a few more points for their consideration (all minor). I congratulate them on completing an interesting body of work!

Figure S2. "although toxicity was abrogated". It would help for the authors to annotate this figure to indicate what the columns represent (presumably serial dilutions of the yeast). It appears that the colonies on the sVIT are smaller than on the full length VIT. Is this consistent with an abrogation of toxicity?

Curve fitting. As mentioned in my previous review, the curve-fitting to the data is often not great, which will impact confidence in calculating EC50 values. In the absence of better curve fits, the authors should specify (either in the methods or the figure legends) what sort of dose-response curves were fitted to the data (e.g. in Fig 2e, 2g, S5a-h, 2g. S3g, 5b)

Line 486. "Fig 8b"

Fig 5d y axis – presumably the units should be min/mg/ml (or min⁻¹ mg⁻¹ ml⁻¹)?

Reviewer #3 (Remarks to the Author):

The authors made satisfactory changes to the manuscript.

I have one minor comment: in multiple occurrences (l613, 615,792...) the authors use the term "promotor" instead of "promoter". Please correct if needed.

Reply to reviewers comments

We thank the reviewers for their comments and have made all the changes requested.

Reviewer #2 (Remarks to the Author):

The authors have addressed my concerns from the previous submission. I have a few more points for their consideration (all minor). I congratulate them on completing an interesting body of work!

Thank you, we are pleased with the corrections and feel that they improve the work, thank you again for all of the comments from the reviewers

Figure S2. “although toxicity was abrogated”. It would help for the authors to annotate this figure to indicate what the columns represent (presumably serial dilutions of the yeast). It appears that the colonies on the sVIT are smaller than on the full length VIT. Is this consistent with an abrogation of toxicity?

We have added the OD600 values to the figure. We agree, this section was badly phrased, we have rephrased it as below to make our meaning clear, the toxicity associated with expression of VIT or sVIT is abrogated in the mutant yeast, rather than a difference between the two constructs.

Expression of full-length VIT in wild type yeast appeared toxic, with fewer and smaller colonies. We also tested an N-terminal truncation (sVIT₆₃₋₃₁₃) as this truncation of PfVIT was shown to successfully complement a $\Delta ccc1$ mutant (Slavic et al., 2016), however sVIT₆₃₋₃₁₃ also showed toxicity in wild type yeast. The toxicity of both VIT constructs appeared abrogated in the $\Delta ccc1$ line, however we did not observe any complementation of the iron-hypersensitivity phenotype.

Curve fitting. As mentioned in my previous review, the curve-fitting to the data is often not great, which will impact confidence in calculating EC50 values. In the absence of better curve fits, the authors should specify (either in the methods or the figure legends) what sort of dose-response curves were fitted to the data (e.g. in Fig 2e, 2g, S5a-h, 2g. S3g, 5b)

Thank you, we have added this information to the Methods section

Line 486. “Fig 8b”

We have corrected this type, thank you

Fig 5d y axis – presumably the units should be min/mg/ml (or min⁻¹ mg⁻¹ ml⁻¹)?

This has been corrected in the figure, thank you for pointing this out

Reviewer #3 (Remarks to the Author):

The authors made satisfactory changes to the manuscript.

I have one minor comment: in multiple occurrences (l613, 615,792...) the authors use the term "promotor" instead of "promoter". Please correct if needed.

Thank you, we have corrected this in the text